# In situ assembly of an injectable cardiac stimulator

Umut Aydemir [1], Abdelrazek H. Mousa [2], Cedric Dicko [3], Xenofon Strakosas [4], Muhammad Anwar Shameem[2], Karin Hellman [1], Amit Singh Yadav[1], Peter Ekström [1], Damien Hughes [1], Fredrik Ek [1], Magnus Berggren [4], Anders Arner[5], Martin Hjort [1] & Roger Olsson [1,2] ✉

Without intervention, cardiac arrhythmias pose a risk of fatality. However, timely intervention can be challenging in environments where transporting a large, heavy defibrillator is impractical, or emergency surgery to implant cardiac stimulation devices is not feasible. Here, we introduce an injectable cardiac stimulator, a syringe loaded with a nanoparticle solution comprising a conductive polymer and a monomer that, upon injection, forms a conductive structure around the heart for cardiac stimulation. Following treatment, the electrode is cleared from the body, eliminating the need for surgical extraction. The mixture adheres to the beating heart in vivo without disrupting its normal rhythm. The electrofunctionalized injectable cardiac stimulator demonstrates a tissue-compatible Young's modulus of 21 kPa and a high conductivity of 55 S/cm. The injected electrode facilitates electrocardiogram measurements, regulates heartbeat in vivo, and rectifies arrhythmia. Conductive functionality is maintained for five consecutive days, and no toxicity is observed at the organism, organ, or cellular levels.

In cardiac arrests or dangerous arrhythmias, a key approach to restoring regular heartbeats encompasses administering electrical stimuli using a defibrillator, an implanted pacemaker, or both. A bioresorbable injectable cardiac stimulator (BICS) administered by a pen-sized device could serve as a minimally invasive and lighter alternative to traditional defibrillators and pacemakers, which necessitate surgery for implantation. A compact BICS would be particularly suited for temporary heart stimulation in remote clinical locations until the patient can be transferred to a facility equipped for permanent implantation. Precise placement of BICS could be guided by imaging, such as ultrasound, similar to the interventional technique of pericardiocentesis[1], but in desperate situations, the device could be guided into position using anatomical landmarks (Supplementary Fig. S1).

In 2021, Rogers and colleagues reported a bioresorbable substrate-bound cardiac pacemaker designed for precise implantation via open surgery[2]. Although the placement of our proposed BICS may be less precise due to its injectable nature, its primary advantage is that open surgery is not required for implantation or removal. Recently, a conductive hydrogel was deployed with precision using a double-barrel syringe via catheter into a cardiac vein. This hydrogel bridged a lesion near the vein in an ablation model, ensuring the maintenance of natural cardiac pacing[3]. The conductivity of the implanted hydrogel was reported to be slightly below 14 mS/cm, approximately twice that of the surrounding tissue (6.0 mS/cm), and in the range of blood conductivity (10–20 mS/cm)[4] Most in vivo injectable conductive hydrogels are employed in a passive capacity (e.g., cardiac patches[5]) or reported without external connectivity and/or low conductivities <22 mS/cm range[6].

[1]Chemical Biology & Therapeutics, Department of Experimental Medical Science, Lund University, Lund, Sweden. [2]Department of Chemistry and Molecular Biology, University of Gothenburg, Gothenburg, Sweden. [3]Pure and Applied Biochemistry, Department of Chemistry, Lund University, Lund, Sweden. [4]Laboratory of Organic Electronics, Department of Science and Technology, Linköping University, Norrköping, Sweden. [5]Department of Clinical Sciences, Lund University, Lund, Sweden. ✉e-mail: roger.olsson@med.lu.se

There is a challenge in designing injectable hydrogels with conductivities high enough to control heartbeats. In addition to applying a high enough bioelectric field for cardiac stimulation, developing injectable electrodes for cardiac applications also faces several other challenges. These include the fact that the formed electrode must firmly adhere to the surface of the beating heart without impairing its natural synchronized movements. It must match the elasticity and stiffness of the cardiac tissue—regardless of the placement of the polymer on the heart. The pro-electrode formulation should be administered through thin needles for minimal invasiveness. Thus, the formulation must be highly soluble for injection but aggregate into a highly conducting structure in vivo to adhere to the beating heart and form an external connection. These critical dualities—adherent vs. elasticity and solubility vs. aggregation—are challenging criteria for material design. Finally, after serving its purpose, the hydrogel should be bioresorbable (cleared from the body) without toxicity.

Given the intricate complexity of developing BICS—criteria that are challenging to assess solely through in vitro methods—we used zebrafish (*Danio rerio*)[7–9]. The in vivo results were further validated using the chicken embryo heart model.

Here, we report on the design, synthesis, and applications of heart monitoring and stimulation of new conductive materials that meet the stringent criteria set for BICS. From a broad range of possibilities, we discovered 8-(2-(2, 5-bis(2, 3-dihydrothieno[3,4-*b*][1,4]dioxin-5-yl)thiophen-3-yl)ethoxy)−1-(trimethylammonio)octane-4-sulfonate (ETE-BuSA) a pseudo-optimal zwitterionic thiophene trimer suitable for in vivo applications. ETE-BuSA mixed with A5, a specific variant of poly(3, 4-ethylenedioxythiophene)butoxy-1-sulfonate (PEDOT-S), formed a highly water-soluble mixture (proBICS). When injected into the pericardial cavity using a small-diameter capillary, proBICS self-organized into a mixed ionic–electronic conducting hydrogel around

the heart. The conductive hydrogel extended out of the pericardial cavity and to the skin surface, creating a contact point to relay an external stimulus. The resulting BICS was used to stimulate the heart. The conductive hydrogel, designed to be temporary, leaves no damage from the electrode. Additionally, animals with this implant showed no behavioral changes during and after its bioresorption, and their offspring exhibited neither developmental nor behavioral issues.

## Results and discussion

### Design and synthesis of materials and characterization

In designing a BICS precursor material (Fig. 1a-i), we anticipated that upon injection into tissue, A5 (Fig. 1a-ii) will integrate ions[10], inducing self-aggregation of A5 and the formation of a conductive framework. Trimers (ETE derivatives) mixed with A5 will diffuse within and out from the initial formed conductive framework. The low oxidation potentials of these trimers (< 0.6 V vs. Ag/AgCl, Supplementary Fig. S2) facilitate electrofunctionalization at low potentials, altering the properties of the A5 backbone and allowing the trimers to attach to the full volume of A5 and form protruding dendrites (Fig. 1b). This method not only positions the conductive framework precisely at the targeted site but also enables the controlled extensions of the conductive structure to interface with the surrounding tissue and cells.

In the search for conductive materials to fit the criteria of BICS, a large number of different materials were evaluated in vivo. The highly dynamic heart region prevented the use of previously explored materials[11], such as the sulfonate (S)- and phosphatidylcholine (PC)-substituted ETE derivatives ETE-S and ETE-PC, which did not generate stable structures on the heart when combined with A5. In fact, the zwitterionic trimer ETE-PC dispersed the A5 core structure before electrofunctionalization. To minimize the dispersion effect observed with ETE-PC while harnessing the favorable toxicity profile of

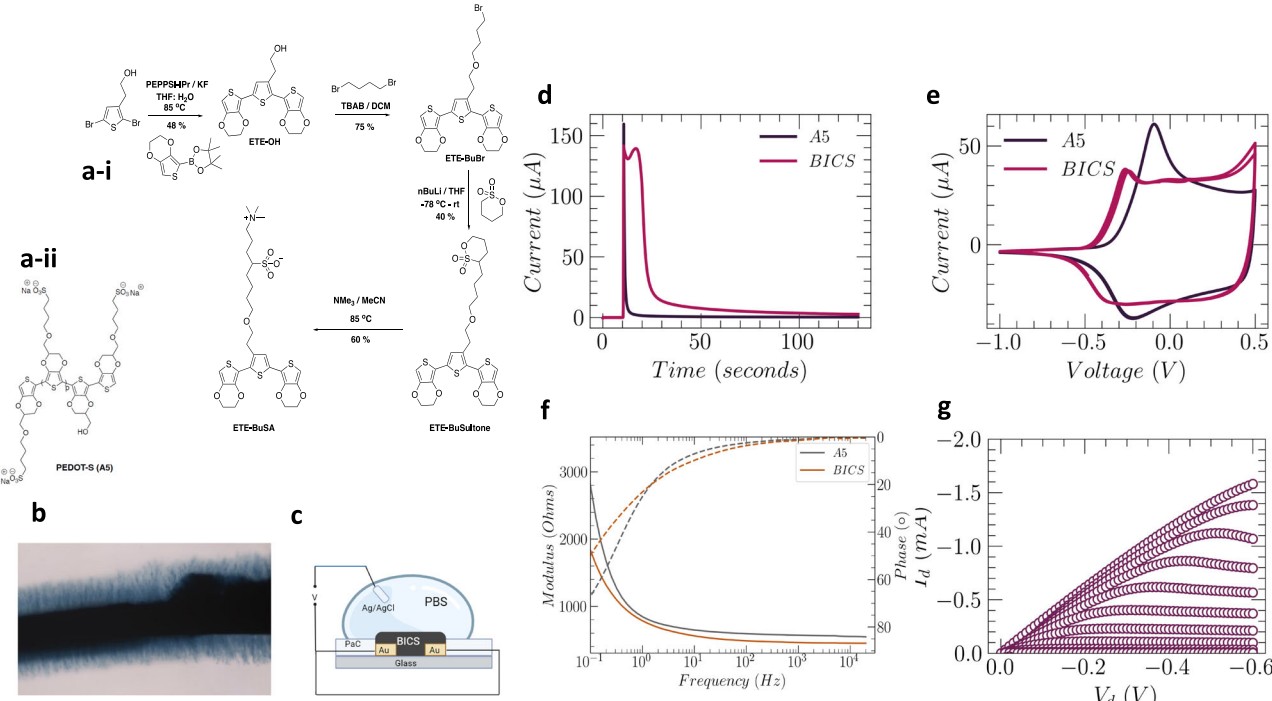

**Fig. 1 | eBICS formation increases performance of A5 and enable dendrite formation. a-i** Synthesis scheme of ETE-BuSA. **a-ii** Chemical structure of A5. **b** BICS after electrofunctionalization (eBICS): a dark core (approximately 100 μm) with protruding dendritic structures is observed. **c** The setup used to characterize the electrical properties of eBICS. **d** Current vs. time measurements during the electrofunctionalization of proBICS. **e** Cyclic voltammetry of eBICS and A5 (average of 3 independent current–voltage measurements). **f** Electrochemical impedance measurements of A5 and BICS. Solid and dashed lines denote modulus and frequency response, respectively. **g** Transfer characteristics of eBICS sweeping the gate voltage from 0 to 0.6 V, with a stepwise increase of 0.05 V. Figure created with BioRender.com released under a Creative Commons Attribution-NonCommcercial-NoDerivs 4.0 International license (https://creativecommons.org/licenses/by-nc-nd/4.0/deed.en).

zwitterions, ETE-BuSA was designed (Fig. 1a-i). ETE-BuSA is a pseudo-optimal zwitterion, one methylene longer between the sulfonate and amine normally used[12]. The mixture of ETE-BuSA and A5 (proBICS) was highly soluble. The positively charged amine interacts with the cell surface—generating a sticky compound—while the sulfonate stabilizes the positive charge on the backbone, contributing to increased doping of the conductive structure.

The synthesis of ETE-BuSA begins by coupling 2-(2,5-dibromothiophen-3-yl)ethan-1-ol with EDOT pinacol boronate ester. This step employed the carbenoid-based palladium catalyst, PEPPSI-IPr, and resulted in the formation of ETE-OH (48% yield). Next, ETE-OH underwent alkylation using dibromobutane in the presence of a tetrabutylammonium bromide (TBAB) as a catalyst, producing ETE-BuBr (75% yield). The synthesis continued by deprotonating 1,4-butane sultone using n-butyllithium (nBuLi) at −78 °C in anhydrous tetrahydrofuran (THF) and quenching with ETE-BuBr, serving as the electrophile, leading to the formation of ETE-BuSultone. Finally, ring-opening of ETE-BuSultone with trimethylamine gave the target molecule, ETE-BuSA. The ETE-BuSA powder was stable at ambient conditions and easy to handle.

When injected into agarose, proBICS formed a dark structure in and around the injection track. Contacting the BICS with external electrodes and supplying 1.0-V allowed ETE-BuSA to functionalize A5 and form a stable gel electrode inside the agarose (eBICS). The formation of eBICS was evident from the darkening of the structure and the formation of dendritic structures extending out from the core structure (Fig. 1b and Supplementary Fig. S12)[11].

The electrical and electrochemical properties of A5, proBICS, and eBICS were characterized on planar metal electrodes with known geometries. proBICS and A5 were drop cast onto the electrodes to form films; the dimensions of the films were defined by peeling off a sacrificial PaC layer (Fig. 1c)[13]. For comparison, known volumes of liquid proBICS and A5 were also electrochemically characterized using the same setup. Figure 1d shows the current as a function of time during electrofunctionalization. For BICS (red line), the non-capacitive current increased for the first 20 s and then dropped rapidly. This drop corresponds to oxidation of the trimer as it is electrofunctionalized in the bulk and surrounding the A5 conductive core. For A5 (black line), the current rapidly decreases to zero, indicating capacitive charging in which ions compensate the holes in the A5 backbone. The oxidation of ETE-BuSA during electrofunctionalization was also followed by absorbance spectroscopy (Supplementary Fig. S3).

eBICS had a specific capacitance of 35 F/cm$^3$, as determined by cyclic voltammetry (Fig. 1e) and electrochemical impedance spectroscopy (EIS) (Fig. 1f). Moreover, compared with A5, the polaron formation peaks for BICS shifted towards more negative values vs Ag/AgCl, indicating a deeper HOMO level and more intrinsically doped conducting polymer (Fig. 1e). BICS was also used as an active material in organic electrochemical transistors (OECTs) (w = 2.4 mm, L = 15 μm, and D = 2.3 μm). The IV curve in Fig. 1g shows a typical transistor behavior for BICS as an OECT channel. eBICS is highly conductive at Vg = 0 V, resulting in a depletion mode OECT, with peak transconductance of $g_m$ = 6.0 mS at Vg = 0.1 V and $V_d$ = −0.6[14]. In addition, the conductivity of electrofunctionalized BICS (55 S/cm) was significantly higher than that of A5 (30 S/cm). The transconductance and specific capacitance of eBICS were comparable to the average values for PEDOT-based OECTs[15,16]. Compared with A5, eBICS was expected to have higher capacitance because functionalization adds trimers inside the A5 matrix, thereby increasing the site density[17].

The elastic properties of eBICS drop cast onto an agarose gel were evaluated by using a mechanical indenter to gently exert a sinusoidal pressure–decompression cycle on the material. The Young's (elastic) modulus of BICS (32 ± 6 kPa) closely matched that of agarose (38 ± 5 kPa) and was several orders of magnitude lower than those reported for PEDOT:PSS hydrogels (2–20 MPa)[17,18]. Moreover, Young's modulus

of eBICS was within the range of human heart tissue (10–200 kPa)[19]. The increase in elastic modulus compared to our previously reported electrodes for brain tissue[11]—that had a shear modulus of 0.57 ± 0.1 kPa compared to 0.5–1 kPa for brain tissue—is probably a function of the additional stickiness needed for the dynamic heart tissue compared to brain tissue and the materials match better with respective tissue. (The statistical analyzes are shown in Data Table S1–4).

## eBICS is biocompatible with cardiac tissue and can control heartbeat frequency ex vivo

To investigate the interaction between BICS and cardiac tissue in isolation, zebrafish hearts were used ex vivo. This approach ensures no interference from other tissues. These hearts remain functional and continue to beat for at least an hour after removal. The application of proBICS to the excised heart and subsequent electrofunctionalization did not induce observable damage (Supplementary Fig. S15). eBICS adhered to the heart, even after extensive rinsing with buffer (Supplementary Movie S5–7). Optical tracking of heartbeats showed that the heartbeat patterns did not differ between hearts without adhered eBICS and hearts with adhered eBICS (Supplementary Movie S1), even when eBICS completely surrounded the heart. These observations indicate that eBICS has a biocompatible functional elastic modulus.

The excised eBICS-covered hearts were stimulated by voltage pulses to alter the beat pattern. eBICS was placed on the atrium of the excised heart and contacted with a Au-coated microelectrode. The counter electrode was placed in the surrounding buffer. Initially, hearts beat at a frequency of 0.8 Hz. Applying voltage pulses stimulated eBICS to control the heart's beating rate, adapting to the stimulating frequency at 2 Hz. After stimulation, heartbeats returned to the original frequency of 0.8 Hz (Supplementary Fig. S14).

## Studies of eBICS in vivo (zebrafish)

To evaluate BICS implantation in zebrafish in vivo, anesthetized zebrafish were placed on a wet sponge, and a micromanipulator was used to insert a proBICS-filled metal-coated microcapillary (30 μm diameter) into the thoracic cavity close to but not into the heart (Supplementary Fig. S16). During retraction of the injection capillary, additional proBICS flowed out to form a continuous structure extending from the pericardium up to the surface of the skin, where a BICS patch was deposited (Fig. 2a). The microinjection capillary (now also used as the anode) was placed on the patch and connected to an external voltage supply to drive the electrofunctionalization at 1 V (counter electrode placed under the fish). After functionalization, where eBICS had formed around the heart, fish were either woken up, kept under anesthesia, or euthanized, depending on which assay to use.

To observe the adherence of eBICS to the heart, the thoracic cavity was surgically opened to expose eBICS. Consistent with the ex vivo zebrafish heart experiment results, eBICS adhered to the heart tissue in vivo and could not be rinsed away. Additional adhesion experiments can be found in (Supplementary Figs. S9–11, Supplementary Movie S5–7, and Data Table S5). The dark eBICS coated part of the heart and formed a continuous track to the surface of the skin, where it created an electrical contact patch (Fig. 2a). After electrofunctionalization, the impedance of the eBICS formed in vivo decreased by an order of magnitude at low frequencies compared with unfunctionalized BICS (Fig. 2b). At high frequencies, the changes in electrode geometry and density in eBICS affected ionic and electronic transport, increasing impedance (Fig. 2b).

To compare the mechanical properties of eBICS on zebrafish hearts with those of cardiac tissue, a small indenter was used to map the eBICS-covered heart in the open thoracic cavity. Static and dynamic indentation measurements were performed on the eBICS-covered hearts and on the hearts of untreated zebrafish. The activity of the heart, mechanically stabilized by the surrounding tissue, was

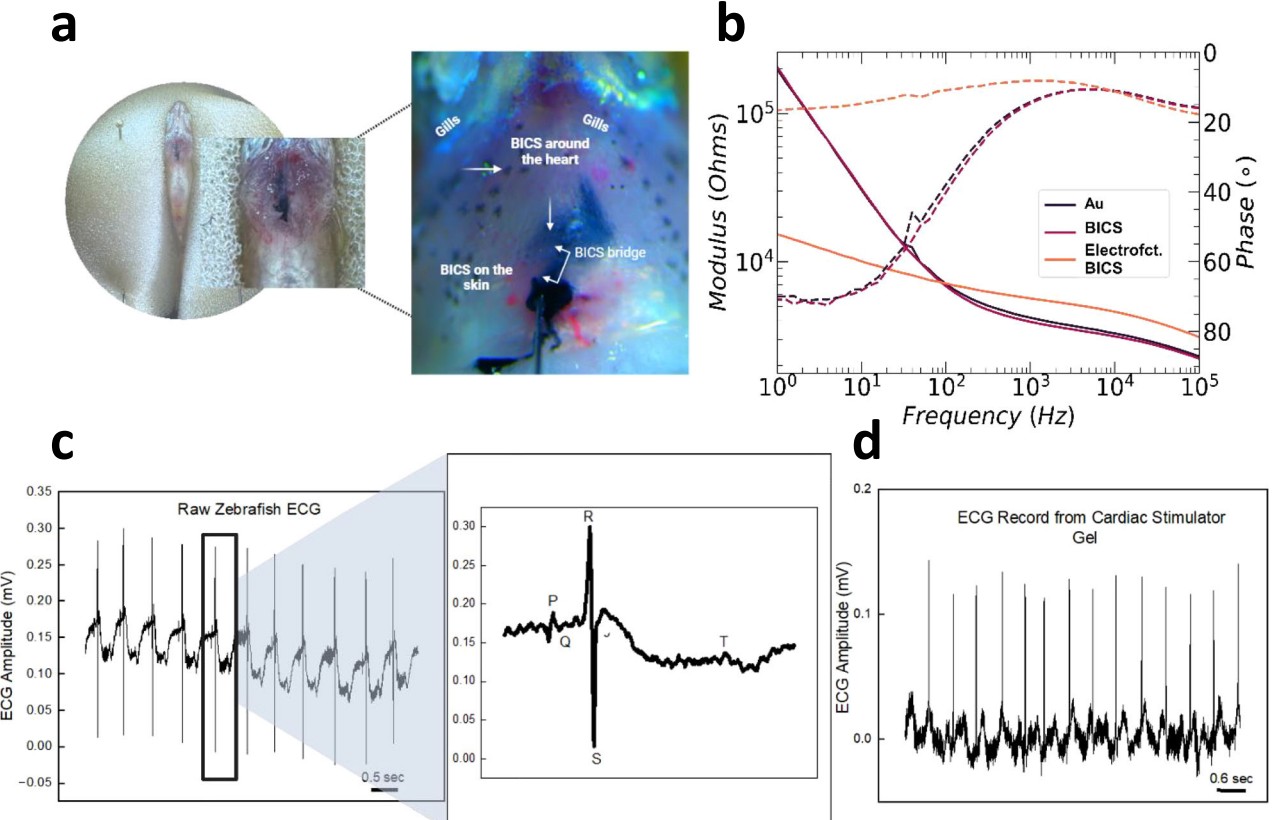

**Fig. 2 | eBICS formed in live zebrafish enable ECG recordings. a** Site of proBICS injection and formation of eBICS in situ. **b** In vivo impedance measurement of BICS and eBICS hearts. **c** The raw ECG signal of zebrafish was recorded within a 2-mV range; the expanded image shows the PQRST complex extracted from the raw ECG signal. **d** ECG signal was recorded via the BICS patch on the zebrafish skin surface. Figure created with BioRender.com released under a Creative Commons Attribution-NonCommcercial-NoDerivs 4.0 International license (https://creativecommons.org/licenses/by-nc-nd/4.0/deed.en).

seemingly unaffected by the small indentations used (100, 200, and 300 μm). Young's modulus, as measured during direct indentation of the eBICS covering the heart, was 21 ± 1 kPa, slightly higher than the Young's modulus of untreated heart tissue (5.4 ± 4 kPa) but still at the low end of the reported range for the human heart[18]. The response upon dynamic stimulation (stress relaxation at 0.1, 1, 1.4, and 2 Hz) did not differ between untreated hearts and eBICS-covered hearts (see Supplementary Material, biomechanical analysis). Here, the dynamic measurement is the most reliable because it corrects for any variability in the indentation rate. Collectively, these results indicate that the mechanical properties of BICS on zebrafish heart are similar to those of cardiac tissue.

The heartbeat patterns of zebrafish injected with eBICS and control zebrafish were compared by recording ECGs with three electrodes. Both eBICS-injected and sham-injected zebrafish were immobilized using tricaine, making direct comparisons possible. The raw ECG signal of zebrafish closely mirrored that of humans; distinct peaks for the P wave, QRS complex, and T wave[20,21] were identified without the need for signal processing (Fig. 2c).

The widespread coverage of the conductive eBICS structure over the heart can result in the P waves being obscured by interference. To gain further insight, we conducted ECG recordings from the eBICS patch on the skin, which extends through the heart surface to the skin. The P waves were identified, indicating minimally perturbative eBICS positioning. To provide a more nuanced understanding of the impact of BICS implantation on heart function, we examined the time interval between successive R peaks (Fig. 2d) to evaluate cardiac rhythm to determine the heart beat frequency. ECG recordings were performed for five consecutive days to assess the long-term functionality of the

BICS patch on the skin, which yielded well-defined ECGs (Supplementary Fig. S4).

Because applying external stimuli interferes with ECG recordings, we could not use ECG to monitor the effect of stimulation on the heartbeat profiles of eBICS-implanted zebrafish. Specifically, stimulation masked low-voltage (0.1 mV) electrical signals from the heart because of a charge and discharge capacitive polymer effect. Due to frequency overlap, attempts to filter out the stimulation pulses were unsuccessful. Thus, seismocardiography, a mechanical method, was used to record the heartbeats of stimulated eBICS-implanted zebrafish[22]. Seismocardiography can track the beating heart over long periods (up to 30 min) and follow changes in beat frequency upon stimulation. A small spherical indenter was used to detect movement at the apex of the thoracic cavity, allowing cardiac movement to be tracked while the stimulating electrodes contacted the eBICS patch. The seismocardiography clearly showed the contractile patterns of the beating heart.

The sedated zebrafish's heartbeat rate was approximately 0.8 Hz, which, during stimulation, increased and adapted to the stimulating frequency of 2 Hz (Fig. 3a, b). Fourier transform shows the non-stimulated frequency and the higher harmonics (Fig. 3c). Control experiments conducted on zebrafish without eBICS showed no change in beating patterns when external stimulation pulses were applied (Supplementary Fig. S5). Interestingly, some zebrafish experienced irregular heart rhythms and arrhythmia, which could be rectified using the eBICS stimulator. Uneven rhythms, extra beats, and missing beats in the spectra were observed; during eBICS stimulation, the arrhythmic heartbeats synchronized and showed no signs of arrhythmia (Fig. 3d,e,f). The setup to record

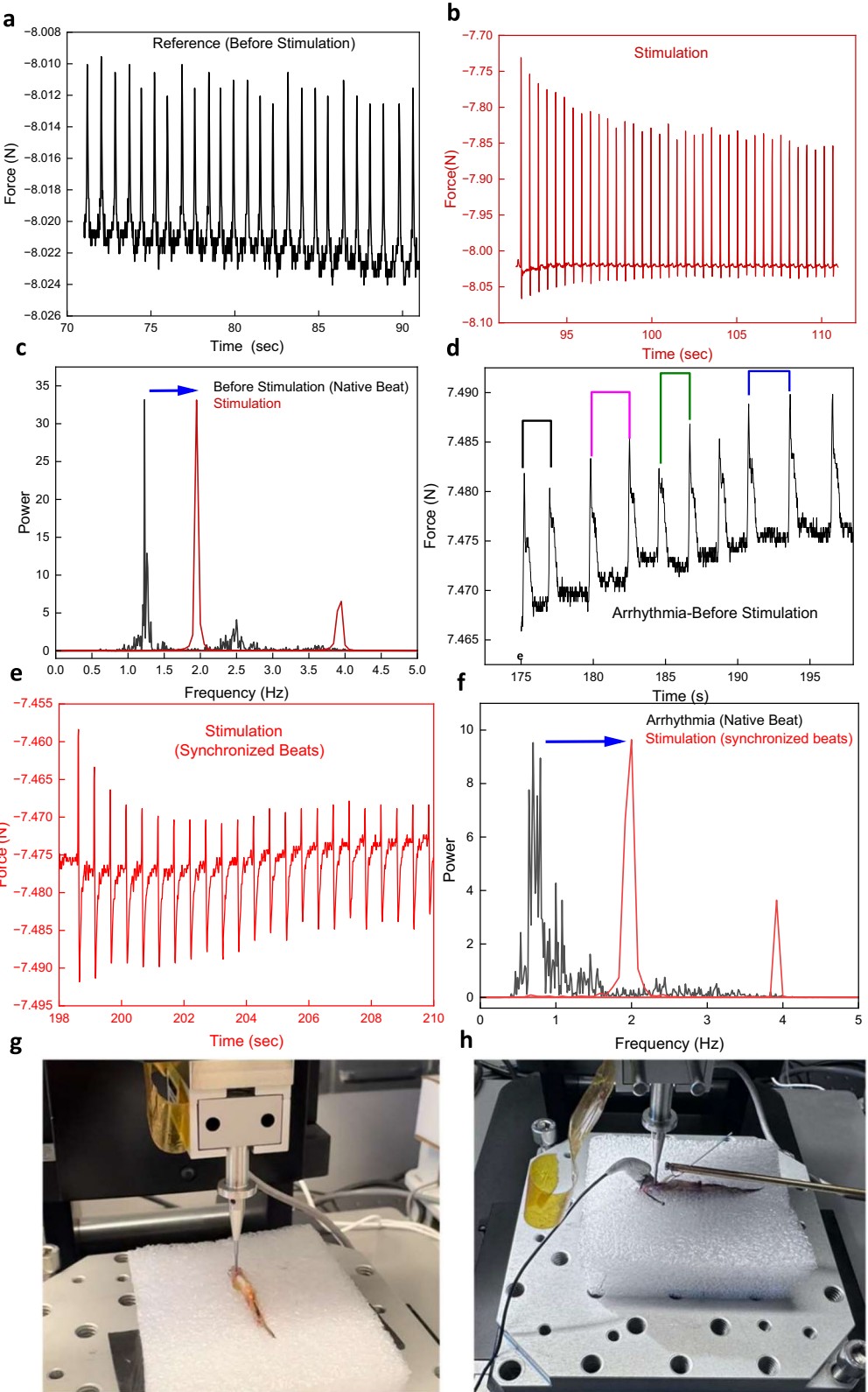

**Fig. 3 | eBICS relaying external stimulation pulses allow control of heart rate and correction of arrythmias.** Seismocardiograms of zebrafish (**a**) before and (**b**) during (in red) external stimulation. **c** Normalized power spectrum of the Fourier transform of the seismocardiograms before and during external stimulation. The blue arrow indicates the shift in the beat frequency. **d** The colored lines indicate irregularities in the beat profile, i.e., arrhythmias, which were resolved by external stimulation (**e**) Synchronized heartbeats during stimulation (in red). **f** The normalized power spectra of the seismocardiograms in (**d**) and (**e**) show the shift to the regular (in red) beat pattern during external stimulation. **g** Seismocardiography setup to record heartbeats with a 1-mm spherical indenter. **h** Recording of external stimulation via seismocardiography.

seismocardiography and recording of stimulation via seismocardiography are presented in Fig. 3g, h.

### eBICS has no long-term physical or behavioral effects on adult zebrafish and their offspring

To determine whether eBICS implantation and resorption have long-term effects on the behavior of implanted zebrafish and their offspring, the behavior of the eBICS-implanted zebrafish was monitored for up to 3 months after proBICS injection. At 1 h post injection (hpi), the zebrafish had recovered and displayed normal behavior (Supplementary Movie S2). The zebrafish were regularly sacrificed up to 35 days post injection (dpi), and eBICS degraded gradually and was completely bioresorbed after 1 week (Supplementary Fig. S4). The zebrafish that were not sacrificed survived for at least 3 months without mortality. During this period, normal behavior was observed in interactions among the eBICS-implanted fish and non-implanted fish. At 21 dpi, the females generated offspring. Anatomical differences were assessed visually at 12 dpf in these larvae, and behavior was assessed in a DanioVision observation chamber. The larvae were exposed to various challenges, such as light and sound, alongside control larvae of the same age bred from the general zebrafish population in the zebrafish facility. No differences in anatomy or behavior, including schooling and shoaling behavior, were observed between BICS-implanted adults and their offspring and control fish.

### eBICS implantation, use, and resorption do not induce inflammation in zebrafish

The tissue toxicity of proBICS injection, heart stimulation, and eBICS bioresorption was evaluated at 1 and 2 hpi and 1, 7, 14, 21, 28, and 35 dpi. At each of these time points, neutrophil accumulation was detected by anti-myeloperoxidase staining; results at 1 hpi, 1 dpi, 2 dpi, and 7 dpi are shown in Fig. 4c–f[23]. No neutrophil accumulation was observed at any time point, including those beyond 7 dpi, indicating a lack of inflammatory immune reaction. These results suggest that the implantation and use of eBICS does not provoke a significant inflammatory response, supporting the biocompatibility of eBICS over short and extended post-injection periods. At present, we do not know the mechanism of bioresorption. However, this is currently addressed in an evaluation of regulatory aspects before moving into human trials. This evaluation will also identify which larger animals are needed to assess function and toxicity simultaneously, aiming to reduce the number of animals used.

### ETE-BuSA and proBICS have minimal cytotoxicity in human fibroblast cells

Following the confirmation of the biocompatibility of eBICS at the organism and tissue levels in zebrafish, toxicity in human cells was evaluated. HFL-1 fibroblast cells were exposed to a series of dilutions of ETE-BuSA alone or proBICS for 24 h, and MTT assays were performed to assess cell viability. Because unfunctionalized molecules quickly diffuse away from the injection site under in vivo conditions, longer time points were unnecessary. Even at extremely high concentrations of ETE-BuSA or proBICS, such as 1000 μg/mL, nearly 60% of the cells remained viable (Fig. 4g). To further validate these findings, the cells were stained with the fluorescent dye calcein-AM, which selectively stains live cells. The results showed that the density of calcein-AM-stained (green fluorescence) cells was not altered by treatment with ETE-BuSA or proBICS at concentrations of up 1000 μg/mL for 24 h (Supplementary Fig. S6). Both toxicity assessments encompassed a broad range of doses, including concentrations of ETE-BuSA up to three times higher than those administered in the in vivo experiments (approximately 240 μg vs. 80 μg). Collectively, these results confirm that ETE-BuSA and proBICS have substantial biocompatibility and minimal cellular toxicity.

### The chicken embryo heart model

The chicken embryo is gaining popularity in cardiology research because its four-chamber structure closely mirrors that of the human heart (Fig. 4h, i)[24]. To obtain chicken heart ECG recordings, chicken embryos were removed from eggs on embryonic development days (EDDs) 14–18. As shown in Fig. 4h, the heart exhibited a fully developed four-chamber structure at this stage. Surface electrodes were placed under the skin, parallel to each other, to record ECGs using the same parameters and setup used previously for zebrafish (Supplementary Fig. S7).

For external electrical stimulation of chicken embryo hearts ex vivo, proBICS was placed on excised hearts to generate eBICS at the atrium (Fig. 4j). Regular heartbeats were observed after eBICS formation, and PQRST complexes were extracted from the ECGs (Supplementary Fig. S7). Thus, eBICS implantation did not negatively impact heart function. After electrofunctionalization, pulses were delivered through the eBICS on the atrium, influencing the beat frequency the same way as in zebrafish. Directly contacting the atrium led to heart stimulation (Supplementary Movies S3 and S4), whereas no stimulation was observed in control experiments in which pulses were applied without direct contact with the heart.

The in vivo experiment from zebrafish was translated to a mouse-sized chicken embryo at EDD 18. The primary challenge involved the manipulation of the chicken embryo; however, a technique involving the lateral opening of the egg facilitated a stable experimental setting. A scaling factor of seven for the materials was found sufficient; this is about one-fourth of what would be estimated using allometric scaling for drugs. The proBICS was administered into the pericardial sac using a 30 G U100-insulin syringe, which was subsequently retracted toward the skin to create the patch for external connectivity. Following this process, BICS was successfully formed on the heart (Fig. 4k). ECGs were obtained through the BICS ($n \geq 10$) and did not show any adverse effect on the heart rhythm (Supplementary Fig. S8).

In this study, we report the design and evaluation of an injectable nanoparticle dispersion that forms a bioresorbable and conductive hydrogel that conforms to the heart surface in vivo. This hydrogel, a mixture of A5 and ETE-BuSA, enables heartbeat monitoring and stimulation as well as the correction of arrhythmia. proBICS can be delivered through a 30-μm syringe needle, demonstrating its high solubility, and self-aggregates into a highly conductive network (BICS) in vivo without external triggers, chemical reactions involving metals[6,25] or dual-barrel delivery systems[3]. Moreover, the growth and functionalization of the conductive network after its initial formation can be controlled to enhance interactions with the biological system—eBICS. The high conductivity and elasticity of the eBICS hydrogel set it apart from other reported hydrogels[26]. Although the use of a small-diameter needle is advantageous for minimally invasive injections, proBICS is also compatible with larger delivery devices used in other research, such as a 400-μm syringe[6] or 7-mm catheter[5]. Thus, the BICS concept opens up an avenue for controlling heart pacing using a compact, low-power system suitable for rural deployment.

## Methods

This study was conducted in accordance with Swedish national legislation and European Community guidelines for animal studies. The ethical committee in Malmö-Lund approved all procedures (5.8.18-05993/2018, 5.8.18-05748/2022, and 5.8.18-19103/2023).

### Synthesis of ETE-BuSA

All commercial chemicals were used as received without further purification. Dry solvents were obtained using an Inert PureSolv MD5 solvent drying system. Column chromatography was performed on a Biotage Isolera instrument using prepacked silica columns (Biotage Snap®), and samples were dry-loaded using Biotage silica. Reactions were carried out under an inert atmosphere of nitrogen and monitored

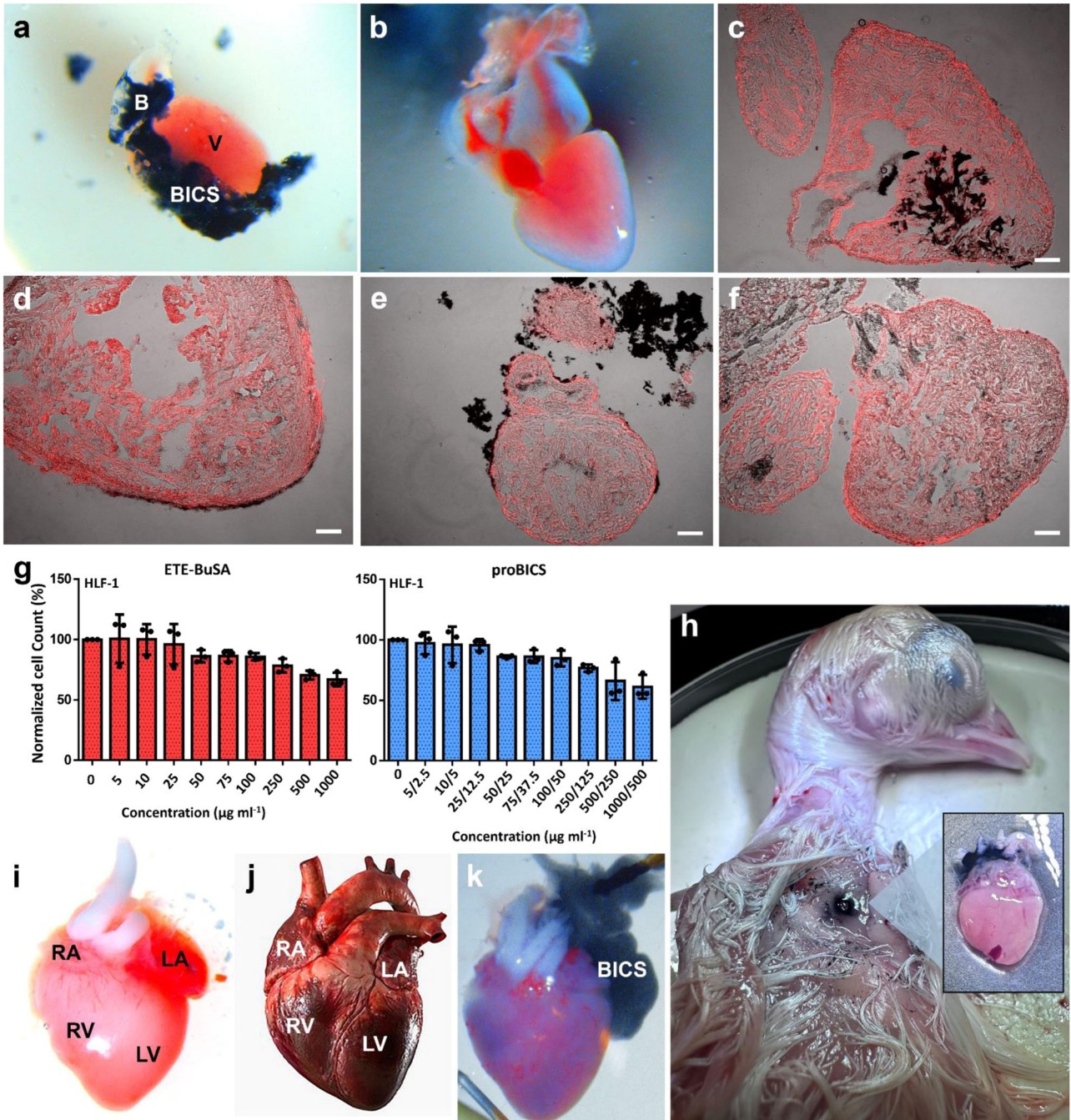

**Fig. 4 | eBICS is resorbed after 7 days and can be applied to larger animals.**
**a**, **b** Dissected hearts of BICS-implanted zebrafish at (**a**) 1 h post-injection (hpi) (B: bulbus arteriosus, V: ventricle) and (**b**) 7 dpi. Complete resorption at 7 dpi is evident from the lack of black BICS hydrogel. (**c**–**f**) Immunohistofluorescence (anti-myeloperoxidase) images of zebrafish hearts at 1 hpi, 1 dpi, 2 dpi, and 7 dpi, respectively ($n \geq 3$ for each time point). Scale bars denote 100 μm. **g** Cell viability of HFL-1 fibroblast cells after treatment with the indicated concentrations of ETE-BuSA or proBICS for 24 h. For proBICS, the concentrations are presented as [ETE-BuSA]/[A5]. ($n = 3$ independent biological replicates each with 3 technical replicates). Untreated cells were used as control. Data are presented as mean values ± SEM." **h**, **i** Comparison of the anatomy of (**h**) the chicken heart and (**i**) a human heart model. The compartments are labeled (RA: right atrium, RV: right ventricle, LA: left atrium, LV: left ventricle). **j** Chicken heart with BICS placed at the atrium. **k** BICS injected chicken embryo, eBICS patch on the skin as and the formation of eBICS around the heart presented (zoomed in picture).

by thin-layer chromatography (TLC, Merck silica gel 60 F254) observed under 254 and 360 nm UV light. ${}^{1}$H and ${}^{13}$C{1H} NMR spectra were recorded on a Bruker Avance III HD 600 MHz or 800 MHz spectrometer. Chemical shifts are given in ppm downfield from TMS. Multiplicities are abbreviated as follows: (s) singlet, (d) doublet, (t) triplet, (q) quartet, and (m) multiplet. High-resolution mass spectra (HRMS) were obtained on an Agilent 1290 Infinity liquid chromatography (LC) system coupled to an Agilent 6520 Accurate Mass quadrupole time-of-flight (Q-TOF) LC/MS with electrospray ionization (ESI).

### 2-(2,5-Bis(2,3-dihydrothieno[3,4-b][1,4]dioxin-5-yl)thiophen-3-yl)ethan-1-ol (ETE-OH)
In a two-neck flask filled with dry THF (50 mL) under a nitrogen atmosphere, 2-(2, 5-dibromothiophen-3yl)ethanol (1.54 g, 5.4 mmol)

was dissolved. To this solution, EDOT boronic ester (3.34 g, 12.5 mmol) was added, followed by the addition of PEPPSI-iPr (0.183 g, 0.27 mmol), and KF (1.87 g, 32.2 mmol). Next, 15 mL of degassed water was added to the reaction mixture and purged with nitrogen for 30 minutes. The mixture was subsequently heated to 85 °C for 6 h. The progress of the reaction was followed by TLC (40 % of EtOAc in pentane). After cooling to ambient temperature, the reaction mixture was filtered through a short silica pad that was subsequently washed with THF and EtOAc. The organic phase was concentrated under reduced pressure, and the residue was purified by column chromatography using a gradient of EtOAc in pentane (0 % → 80 %), which yielded a yellow foam (1.05 g, 48 % yield) [1]H NMR (600 MHz, CD$_3$CN) δ 7.12 (s, 1H), 6.46 (s, 1H), 6.32 (s, 1H), 4.34–4.30 (m, 2H), 4.27 (ddd, $J$ = 5.5, 3.1, 1.2 Hz, 2H), 4.23 (tdd, $J$ = 3.9, 3.3, 2.1 Hz, 4H), 3.70 (td, $J$ = 6.9, 5.6 Hz, 2H), 2.85 (t, $J$ = 6.9 Hz, 2H), 2.72 (t, $J$ = 5.7 Hz, 1H).[13]C NMR (151 MHz, CD$_3$CN) δ 143.2, 142.9, 139.6, 139.1, 138.4, 134.6, 127.7, 126.1, 112.1, 110.1, 100.0, 97.9, 66.1, 65.9, 65.6, 65.5, 62.6, 33.6.

### 5, 5′-(3-(2-(4-Bromobutoxy)ethyl)thiophene-2, 5-diyl)bis(2,3-dihydrothieno[3, 4-b][1,4]dioxine) (ETE-BuBr)

ETE-BuBr was synthesized according to a published procedure[27]. A solution of ETE-OH (1.0 g, 2.45 mmol, 1 equiv.) in 20 mL DCM was added to a mixture of 1,4-dibromobutane (3.5 mL, 29.3 mmol, 12 equiv.) and tetrabutylammonium bromide (TBAB) (239 mg, 0.74 mmol, 0.3 equiv.) in 10 mL of DCM. This mixture was stirred for 15 min before adding of 30 mL of an aqueous 50 wt % NaOH solution. The resulting biphasic system was then stirred vigorously overnight, and the progress of the reaction was followed by TLC (40 % of EtOAc in pentane). The reaction mixture was diluted with 100 mL of water, and the product was extracted using three 100 mL portions of DCM. The organic layer was dried over anhydrous Na$_2$SO$_4$, filtered, and the solvent was removed under reduced pressure. The residue was purified by column chromatography using a gradient of EtOAc in pentane (0 % → 40 %) to yield the product as a sticky yellow liquid (1 g, 75 % yield). [1]H NMR (600 MHz, CD$_3$CN) δ 7.12 (s, 1H), 6.43 (s, 1H), 6.29 (s, 1H), 4.32–4.29 (m, 2H), 4.24 (td, $J$ = 3.6, 2.0 Hz, 2H), 4.21 (ddt, $J$ = 6.3, 4.0, 2.3 Hz, 4H), 3.58 (t, $J$ = 6.7 Hz, 2H), 3.44 (t, $J$ = 6.8 Hz, 2H), 3.40 (t, $J$ = 6.2 Hz, 2H), 2.87 (t, $J$ = 6.7 Hz, 2H), 1.90–1.81 (m, 2H), 1.65–1.58 (m, 2H).[13]C NMR (151 MHz, CD$_3$CN) δ 143.15, 142.86, 139.57, 139.01, 138.34, 134.56, 127.61, 126.15, 112.14, 110.06, 100.01, 97.92, 70.96, 70.36, 66.07, 65.84, 65.55, 65.44, 35.27, 30.67, 30.52, 29.03.

### 3-(4-(2-(2, 5-Bis(2, 3-dihydrothieno[3, 4-b][1,4]dioxin-5-yl)thio-phen-3-yl)ethoxy)butyl)-1, 2-oxathiane 2, 2-dioxide (ETE-BuSultone)

n-BuLi (2.5 M in hexanes, 1.85 mmol, 740 µL) was added dropwise to a solution of 1,4-butane sultone (170 µL, 1.66 mmol) in 6 mL of dry THF at −78 °C under an inert nitrogen atmosphere. The mixture was stirred at −78 °C for 30 minutes. Next, a solution of ETE-BuBr (897 mg, 1.65 mmol) in 6 mL of dry THF was added, and a pale red solution formed immediately. The reaction mixture was stirred at −78 °C for 30 min; the cooling bath was then removed, and the reaction was allowed to proceed at ambient temperature overnight. The reaction was quenched using a small amount of water, and the solvents were removed under reduced pressure. The residue was redissolved in 100 mL of water and extracted with two 100 mL portions of EtOAc. The organic phase was dried over anhydrous Na$_2$SO$_4$ and filtered, and the solvent was removed under reduced pressure. The crude product was purified by column chromatography using a gradient of EtOAc in pentane (0 % → 100 %). The title compound was obtained as a yellow foam (393 mg, 40 % yield). [1]H NMR (600 MHz, CD$_3$CN) δ 7.13 (s, 1H), 6.46 (s, 1H), 6.32 (s, 1H), 4.42 (ddd, $J$ = 8.9, 3.4, 1.7 Hz, 2H), 4.34–4.31 (m, 2H), 4.26 (ddd, $J$ = 5.4, 3.1, 1.1 Hz, 2H), 4.25–4.20 (m, 4H), 3.60 (t, $J$ = 6.7 Hz, 2H), 3.40 (t, $J$ = 6.0 Hz, 2H), 3.06 (dddd, $J$ = 11.1, 7.7, 5.7, 3.8 Hz, 1H), 2.89 (t, $J$ = 6.7 Hz, 2H), 1.92–1.71 (m, 4H), 1.57–1.44 (m, 4H), 1.44–1.36 (m, 1H).

### 8-(2-(2,5-Bis(2,3-dihydrothieno[3,4-b][1,4]dioxin-5-yl)thiophen-3-yl)ethoxy)-1-(trimethylammonio)octane-4-sulfonate (ETE-BuSA)

A solution of 4.2 M trimethylamine (1.3 mL, 5.46 mmol, 14.8 equiv.) in ethanol was added to a degassed solution of ETE-BuSultone (0.220 g, 0.37 mmol, 1 equiv.) in 1.5 mL of dry acetonitrile in a 15-mL pressure tube. The tube was heated to 85 °C overnight. After cooling to ambient temperature, the crude solution was transferred to a small flask, and the solvents were removed under reduced pressure, yielding the title compound as a pale-yellow foam in a sticky yellow liquid. The residue was purified by column chromatography (EtOAc:MeCN:MeOH:H$_2$O 3:1:1:1) to obtain a fluffy yellow solid (145 mg, 60% yield). [1]H NMR (800 MHz, CD$_3$OD) δ 7.15 (s, 1H), 6.46 (s, 1H), 6.32 (s, 1H), 4.37–4.32 (m, 2H), 4.30–4.26 (m, 2H), 4.26–4.21 (m, 4H), 3.65 (td, $J$ = 6.7, 3.1 Hz, 2H), 3.48 (t, $J$ = 6.0 Hz, 2H), 3.28 (dt, $J$ = 12.2, 6.0 Hz, 1H), 3.20 (dd, $J$ = 12.1, 5.4 Hz, 1H), 3.07 (s, 9H), 2.91 (t, $J$ = 6.7 Hz, 2H), 2.68 (tt, $J$ = 8.1, 4.2 Hz, 1H), 1.98 (tdd, $J$ = 15.4, 8.3, 4.4 Hz, 3H), 1.75 (dddd, $J$ = 14.7, 9.4, 7.4, 5.5 Hz, 1H), 1.69–1.60 (m, 2H), 1.60–1.52 (m, 3H), 1.49–1.43 (m, 1H).[13]C NMR (201 MHz, CD$_3$OD) δ 143.59, 143.26, 139.77, 139.20, 138.37, 135.22, 128.01, 126.24, 112.80, 110.62, 100.15, 97.84, 71.49, 67.77, 66.42, 66.17, 65.89, 65.76, 60.21, 53.56, 31.16, 30.78, 30.74, 27.60, 24.84, 21.41. HRMS (ESI) m/z: [M + H]$^+$ calculated for C$_{29}$H$_{40}$NO$_8$S$_4$: 658.1637, found: 658.1649.

## Preparation of proBICS

ETE-BuSA was dissolved at 40 mg/mL in milli-Q water. A5 (a variant of PEDOT-S)[10,11] was added to this solution to achieve a final A5 concentration of 20 mg/mL. This mixture is denoted as the bioresorbable injectable cardiac stimulator solution (proBICS); the conductive structure formed after injection (BICS), and after electro-functionalization of BICS (eBICS, for method, see below). The term 'bioresorbable' is used because the electrode is spontaneously absorbed by the body, leaving no trace and without exerting toxicity. However, the mechanism by which this occurs is currently unknown.

To ensure a well-dispersed solution, the proBICS was ultra-sonicated before loading into microinjection capillaries.

## In vitro electrical measurements

Custom-designed gold (Au) electrode arrays were used for in vitro electrical measurements following protocols outlined by Strakosas et al. [13]. In short, gold electrodes were patterned on glass slides by UV lithography, and the slides were subsequently coated with parylene C (PaC) for use with a peel-off technique to expose the contacts and the interdigitated array. proBICS was drop cast onto the electrodes and either dried under N$_2$ gas or maintained as a liquid. Two consecutive Au lines with a length of 2.4 mm, width of 15 µm, and distance between them of 15 µm were electrically biased simultaneously (Keithley 2612 A SourceMeter), and the resulting current was registered. To determine the conductivity of the BICS, four-probe or two-probe current–voltage measurements were performed at several different electrode distances, and the transmission line model was used.

## Microinjections into the pericardium

Zebrafish were anesthetized in 1.5 mM tricaine (ethyl 3-aminobenzoate methanesulfonate) (Sigma Aldrich, E10521) until fish had no opercular movements and did not react to tail fin pinching. The anesthetized fish were placed in a sponge and positioned in the microinjection setup. Scales (2–4, depending on the size of the fish) were removed from the skin just above the heart, and a small hole was made in the skin with a 30 G 0.3 × 13 mm Terumo Agani™ Needle (AN*3013R1). The metal-coated glass capillaries used for injections had a beveled tip with a diameter of 30 µm (BM100T-15, beveled, straight, shortened, and fire-polished ends from BioMedical Instruments GmbH). One side of the capillary was coated in-house with 50 nm palladium/platinum (Pd/Pt) alloy using a sputterer (Quorum, QT 150). The capillary was loaded

with 2 μL of proBICS using a 20 μL microloader (epT.I.P.S., Eppendorf). For pericardial microinjection, a micromanipulator (Quad Sutter Instrument) was used to insert the proBICS loaded capillary through the previously created hole, parallel to the sagittal plane, into the pericardium. Once the capillary reached the pericardium, a single injection was applied with a balance pressure of 1.8 psi and injection pressure of 8.0 psi for 10 ms (Warner PLI-100A) The balance pressure ensured a continuous material flow while retracting the capillary, creating a BICS patch to the skin surface where additional proBICS was deposited to leave a patch. This patch was used as the connection point for delivering external stimulation.

## Ex vivo stimulation of excised hearts

Adult wild-type (AB) zebrafish were euthanized by immersion in an ice-cold water bath for 15 min, and their hearts were dissected. The beating hearts were maintained in a petri dish filled with Ringer's solution at a temperature of 27 °C. A 2 μL volume of proBICS was placed in the solution adjacent to the heart to create an electrode connected to the atrium. A 25 μm Au-coated W-microelectrode (Signatone, Gilroy, CA) was used to contact the BICS (anode), and the counter electrode was positioned in the surrounding Ringer's solution. During the electrofunctionalization of BICS, a 1.0-V bias was applied for 5 min using a Keithley SourceMeter 2612 A (Keithley Instruments), forming eBICS. The eBICS was then used to relay external electrical signals. The eBICS was used as the cathode, and the counter electrode was positioned in the surrounding Ringer's solution. The electrical stimulation was carried out using a Grass S48 Stimulator (AstroMed) supplying square voltage pulses with an amplitude of 4 V and pulse width of 10 ms.

For harvesting chicken hearts, the fertilized eggs from domestic chickens (*Gallus gallus*) were washed with 70% ethanol and incubated at 37.5 °C and 60% humidity. The eggs were turned for 30 s twelve times daily in an egg incubator (Chicti). On embryonic development day (EDD) 4, approximately 4–6 mL of albumin was withdrawn through the shell on the side of each egg using a 20 G syringe. The eggshell on the blunt side was removed with a stainless-steel egg topper followed by a small scalpel blade. The opening was covered with a 35 mm petri dish lid, and the egg was placed in an incubator (Heka-Brutgeräte) at 37.5 °C and 60% humidity until EDD 14 and by this stage, they were euthanized by decapitation. The hearts were dissected from the body and transferred to a petri dish containing 37 °C Howard Ringer's solution. 5 μL of proBICS solution was placed, electrofunctionalized and the hearts externally stimulated with the same paramethers mentioned for ex vivo zebrafish heart stimulation.

## Ex vivo ECG recording of chicken embryo hearts

For ex vivo ECG recording, EDD 14 embryos were euthanized by decapitation, and the hearts were dissected and transferred to a petri dish containing 37 °C Howard Ringer's solution. Stainless steel sub-dermal electrodes (12 mm long, 29 G) were positioned in the liquid adjacent to the heart, and electrocardiograms (ECGs) were recorded with an AD Instruments Bio Amplifier in conjunction with a PowerLab 4/20 data acquisition system and Chart 4.2 software. The recording was performed in the 2-mV range with a 50 Hz notch filter, a 0.3 Hz high-pass filter, and a 1 kHz low-pass filter to ensure accurate signal processing[20].

## In vivo stimulation of zebrafish hearts

Adult wild-type (AB) zebrafish were implanted with eBICS as described previously, and electrofunctionalized with 1.0 V from the BICS patch on the skin surface with the injection capillary (used as anode). The electrical stimulation was performed using 4.0 V square-voltage pulses with a 10-ms pulse width supplied by a Grass S48 Stimulator (AstroMed).

## In vivo electrical measurements of impedance

Adult wild-type (AB) zebrafish were implanted with eBICS, as described previously. For impedance potentiometry, a small incision was made in the anesthetized zebrafish to expose the heart. The working electrode was placed close to the injection track, reaching the heart surface covered by eBICS, and a Ag/AgCl counter electrode was positioned on the skin of the fish at a distance from the injection. Impedance was measured using a potentiostat (Gamry Instruments) and Nova 2.1.6 software to record data. Measurements were performed in a range of 1–10$^5$ Hz[13]. As a control, the same procedure was performed using zebrafish that had not been injected with proBICS.

## In vivo surface ECG recording in adult zebrafish and chicken embryos

ECGs were recorded following the procedures as outlined by Nguyen[20]. For zebrafish, subdermal electrodes were positioned on the body's surface, near the cardiac region. The electrical signals originating from the zebrafish heart were acquired and amplified using an AD Instruments Bio Amplifier in conjunction with a PowerLab 4/20 data acquisition system and Chart 4.2 software. ECG signals were continuously monitored and recorded in the 2 mV range with a 50 Hz notch filter, 0.3 Hz high-pass filter, and 1 kHz low-pass filter to ensure optimal signal processing. To investigate the effect of eBICS on cardiac properties, adult wild-type (AB) zebrafish were implanted with eBICS as described previously. The ECG electrodes were placed above the injection site, parallel to each other, to enhance the signal-to-noise ratio. For ECG recordings through eBICS, the positive electrode was positioned on the connection patch to eBICS, while the negative electrode was positioned at the lower part of the heart, and the ground electrode was clipped onto the tailfin.

To obtain in vivo surface ECGs of chicken embryos, EDD 14 or older embryos were transferred to a wet sponge hydrated with 37 °C Howard's chicken Ringer saline at pH 7. The anode was placed under the skin by the atrium, and the cathode was placed under the skin at the ventricle. The ground electrode was clipped to the embryo's legs. The settings used to record the ECGs were the same as those used for zebrafish.

## Injections to chicken embryos

EDD 18 chicken embryos were exposed by cutting the eggshell from blunt side top to bottom laterally, while care was taken to keep embryos within the fluid surrounding. 15 μL of BICS was injected into pericardial sac of embryos with 30 G U100-insulin syringes (BD Micro-FineTM+Demi) and a continuous injection was performed while retracting the syringe to form a BICS patch on the skin. Electro-functionalization was performed using 1.0 V bias for 5 minutes by using a Keithley Sourcemeter with a Ag/AgCl electrode contacted to the BICS patch. After injections, ECG was recorded with the same procedure mentioned above.

## Biomechanical performance characteristics of eBICS

The biomechanical performance testing and mechanical compatibility of the eBICS were assessed by indentation tests (DIN EN ISO 14577-1:2015) using a Mach−1 Model V500css testing machine (Biomomentum Inc., Laval, Canada). For in vitro measurements, 2 μL of proBICS was dropped on the surface of an agarose block (0.5% in Ringer buffer), and electrofunctionalized using a 1.0 V bias supplied by a Keithley 2612 A SourceMeter (Keithley Instruments) for 5 min. For in vivo measurements, zebrafish were injected with 2 μL of proBICS followed by electrofunctionalization as described previously. After eBICS implantation, the zebrafish were euthanized in ice water. The thorax was opened without disrupting the eBICS, and measurements were taken on the heart in the cavity, placing the mechanical indenter directly on the eBICS covering the heart.

The load cell was nominally 0.1 N (10 gram force [gf]) with a resolution of 75 μN. Similar to the seismocardiography measurements, a 1 mm spherical indenter was gently brought into contact with the heart or polymer. The contact conditions were optimized to an indenter speed of 0.05 mm/s and a stop force condition of 0.5 gf (approximately 5 mN). The system was allowed to relax (i.e., no residual force) for at least 10 min. Following the wait time, a 3-ramp stress-relaxation sequence was applied at an indenter speed of 0.01 mm/s and indentation depth of 100, 200, and 300 μm. For each of the ramps, the heart or polymer was indented and allowed to relax until the force did not change. After the stress relaxation, the heart or polymer was dynamically excited at frequencies of 0.1, 1, 1.4, and 2 Hz with an amplitude of 50 μm. These excitation frequencies were chosen based on biomechanical data on the native zebrafish heartbeat profile[8,9]. The resulting mechanical data were analyzed according to the methods described in refs. 28,29.

### Seismocardiography

For seismocardiography, a Mach-1 Model V500css testing machine (Biomomentum Inc., Laval, Canada) was used to record mechanical movements in an indentation on the surface of the zebrafish. The load cell was nominally 0.1 N (10 gram force [gf]) with a resolution of 75 μN. An anesthetized zebrafish was placed on a wet sponge, and a 1-mm spherical indenter was gently brought into contact with the top of the thoracic cavity. The contact conditions were optimized to an indenter speed of 0.05 mm/s and a stop force condition of 0.5 gf (~5 mN). The seismocardiogram was recorded continuously using Mach-1 Motion Software (Version 4.4.0.9). The power spectrum of the seismocardiogram was computed to extract dominant frequencies in the signal.

### Immunofluorescence during and after bioresorption of eBICS

To detect neutrophils as an indicator of inflammatory activity, anti-myeloperoxidase antibody staining was performed at 1 and 2 h post-injection (hpi) and 1, 7, 14, 21, 28, and 35 days post injection (dpi). Excised zebrafish hearts were fixed overnight in 2 % paraformaldehyde (PFA) (4% Histolab in 0.1 M phosphate buffer HL 96753.1000, diluted 1:1 with phosphate buffer). Next, the hearts were rinsed with phosphate-buffered saline (PBS) pH 7.4, cryoprotected in 25 % (w/v) sucrose (Sigma, > 99.5 % BioXtra S7903) in PBS, frozen in NEG 50 ™ OCT (Epredia, 6502) on dry ice and cryosectioned into 20 μm-thick sections. The sections were rinsed three times for 3 min each with PBS, including 0.25 % Triton X (Sigma, X-100) (PBS-TX) at pH 7.2. After blocking in 10 % goat normal serum in PBS-TX for 2 h at room temperature, the sections were incubated overnight at +4 °C with myeloperoxidase antibody (1:500; Abcam, #ab210563, rabbit polyclonal to myeloperoxidase, LOT# GR3443812-4) in PBS, including 0.25 % Triton X with 1 % Native Cow Bovine Serum Albumin protein (Abcam, #ab64009) (PBS-TX-BSA).

The sections were then rinsed three times in PBS-TX for 5 min each and incubated with goat anti-rabbit IG Alexa Fluor 546 conjugate (1:200 in PBS-TX-BSA; Invitrogen, LOT # 2539808) for 2 h. Then, after three rinses in PBS, the sections were mounted on coverslips using ProLong™ Gold antifade reagent with DAPI (Invitrogen, #P36931) as the mounting medium. To analyze the histologically stained sections, an Olympus IX73 microscope with a Hamamatsu Orca R² camera was used. At least three zebrafish were used at each time point, and a total of 30 zebrafish were examined.

### Behavioral studies of adult zebrafish and their offspring

Zebrafish embryos were raised under a 14 h:10 h light:dark cycle at 28.5 °C. The embryos were kept in Petri dishes containing E3 embryo medium (5 mM NaCl, 0.17 mM KCl, 0.33 mM CaCl₂, and 0.33 mM MgSO₄) in an incubator until five days post-fertilization (dpf). The larvae were then transferred to a 0.8-L aquarium, which was placed in a recirculating system maintained at a temperature of 26 ± 1.5 °C (Aquaneering, Inc., San Diego, CA).

At this stage, feeding with a commercial larval diet (ZM000, ZM Fish Food & Equipment, Winchester, UK) three times daily was initiated. Behavioral experiments were conducted at 12 dpf. A DanioVision observation chamber was used (Noldus Information Technology, Netherlands). Twelve larval offspring of the proBICS-injected, eBICS-implanted, and stimulated group of zebrafish were individually allocated to the first and third rows of a 4 × 6 well plate, with each well containing one larva. The control group of larval offspring from untreated zebrafish was individually allocated to the second and fourth rows of the same plate. The larvae were allowed to adapt to the experimental conditions at 27 ± 1.0 °C for 1 h. The behavior analysis spanned 50 min in the dark and included two periodic challenges, white light and tapping, which were repeated four times throughout the day. The video recordings were analyzed using EthoVision® XT 15 software (Noldus Information Technology, Wageningen, Netherlands) to assess parameters such as distance moved, acceleration, velocity, meandering, and rotation. Zebrafish (adults and offspring) were observed for 3 months (the longest time period allowed by the ethics permit) to evaluate behavioral and/or anatomical differences.

### Cell culture and cytotoxicity assay

The human lung fibroblast cell line HFL-1 (ATCC #CCL-153) was cultured in Dulbecco's modified Eagle's medium (DMEM) (Gibco) with 10% fetal bovine serum (FBS) (Gibco), 100 units of penicillin, 100 μg/ mL of streptomycin, and 1% non-essential amino acids in an incubator at 37 °C with 5% $CO_2$. To assess the toxicity of ETE-BuSA and proBICS, the MTT assay was performed. Cells were seeded at $2 \times 10^4$ cells per well in a flat-bottom 96-well plate and allowed to attach/ habitutate for one day. The cells were then treated with ETE-BuSA (5, 10, 25, 50, 75, 100, 250, 500, and 1000 μg/ml) and proBICS (2:1 ratio of ETE-BuSA:A5; effective concentration of 5:2.5, 10:5, 25:12.5, 50:25, 75:37.5, 100:50, 250:125, 500:250 and 1000:500 μg/ml respectively) for 24 h. Next, 200 μL of 0.5 mg/mL MTT (Merck, Cat no. M2128) was added to each well and incubated at 37 °C for 4 h. Finally, 200 μL of isopropanol (Merck, Cat no. 34863) was added to dissolve the formazan crystals, and the optical density of the formazan solution (a measure of live cells) was determined in a microplate reader at 570 nm (Spark Cyto, Tecan). The formazan signal (live cell count) was normalized to that of the untreated control samples. Three biological and three technical replicates were performed for each condition. In a separate experiment, cells were treated with ETE-BuSA alone or proBICS, as described above, followed by the live cell staining dye calcein-AM (2.0 μM) (Thermo Fisher, Cat no. C3100MP) for 30 min at 37 °C. Fluorescence images of the cells were acquired with a 4x objective using a multi-mode plate reader with fluorescence imaging function (Spark Cyto, Tecan).

### Statistics and reproducibility

Sample size was not predetermined by any statistical method; however, the number of animals used in the study was minimized. The study adhered to $n \geq 3$ biological replicates for each experiments. For electrical measurements, electrically and optically verified short-circuited contacts were excluded from the analysis. One-way analysis of variance (ANOVA) was used for the biomechanical analysis of the heart and heart/eBICS interactions.

### Reporting summary

Further information on research design is available in the Nature Portfolio Reporting Summary linked to this article.

## Data availability

All data supporting the findings of this study are available within the article and its supplementary files. Any additional requests for information can be directed to, and will be fulfilled by, the corresponding authors. Source data are provided in this paper.

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

## Acknowledgements

This study was accomplished within the Lund University Strategic Research Areas MultiPark and NanoLund. We thank Dr. Sara Rolandsson-Enes (Lund University) for donating HFL-1 cells and Marios Savvakis, and Mary J Donahue for providing expertize on device fabrication for electrical measurements. This research was enabled by equipment within the Lund Nano Lab (LNL) and Lund University Bioimaging Center (LBIC). We acknowledge funding from the Swedish Research Council (2023-04965, R.O, and 2021-05231, M.H.), the Swedish Foundation for Strategic Research (RMX18-0083, M.B. and R.O.), and the European Research Council (AdG 2018, 834677, M.B.). Additional funding was provided by the Swedish Research Council (R.O.) and a Novo Nordisk Foundation Distinguished Innovator Grant (0087092, R.O.).

## Author contributions

Conceptualization: R.O. Methodology: R.O., and A.A. Design and synthesis: R.O., A.H.M., and M.A.S. Zebrafish experiments: U.A., K.H., A.A., P.E., D.H., and F.E. Chicken experiments: U.A. and K.H Cell work: A.S.Y. Electrical characterization: U.A., X.S. Mechanical characterization: U.A., C.D., and D.H. Funding acquisition: M.B., M.H., R.O. Supervision: R.O. Writing – original draft: U.A., A.H.M., M.H., and R.O. Writing – review & editing: All authors reviewed and edited the draft.

## Funding

## Competing interests

The authors declare no competing interests.
