## [Peer Review File · Nature Communications]

REVIEWER COMMENTS

Reviewer #1 (Remarks to the Author):

This paper presents an injectable conductive hydrogel as an electrode for the cardiac application, including both ECG monitoring and electrical stimulation. The material consists of A5 (a variant of PEDOT-S) and ETE-Busa and has low Young's modulus (21kPa) with high conductivity (55 S/m). The authors demonstrate the efficacy of this material by using zebrafish and chicken embryo.

1. We recommend that the authors provide additional data to support their claim – (1) bioresorbability. Although the title of this manuscript is “bioresorbable injectable cardiac stimulator”, data that support the bioresorbability of the materials is insufficient. The authors have to provide relevant data. In addition, thiophene, in its basic form, is not inherently bioresorbable due to their stable aromatic ring structure, and this is the reason why PEDOT series are known as biocompatible but non-degradable (and non-bioresorbable) materials. In principle, bioresorbable materials refer to the materials those can be broken down and absorbed by the body, usually through natural biological processes. In other words, if it is bioresorbable, it should be hydrolysable or can be broken by the enzyme in our body. Therefore, the authors should demonstrate the bioresorbability of the synthesized material.

2. The authors should add more data regarding the biocompatibility and toxicity of the synthesized material and of its degradation byproduct. Histological examination of myocardial tissue near the site of material attachment, immunohistochemistry should be conducted.

3. We recommend the authors revise the title since it can mislead the readers. What the authors developed is just electrode, and it is not a cardiac stimulator. Cardiac stimulator generally refer to the pacemaker with battery and controlling system, but this system should be connected with external stimulator (or ECG sensor) to be operated.

4. The authors should add more information to support their claim – (2) the necessity of the higher conductivity of the electrode. As the authors mentioned, injectable hydrogel is already demonstrated in many papers, and they show reliable stimulation (or pacing) performance. The authors should explain the advantages of higher conductivity and demonstrate it in the main manuscript with comparison data. It is also recommended to provide a table that compares the various properties, including electrical conductivity, adhesiveness, degradability, viscosity etc., of published hydrogels and developed material.

5. The authors need to do more literature review for the modulus of reported hydrogels. Although the authors highlight the low modulus of the sample with that of PEDOT:PSS hydrogel (Page 7, line 174), it is commonly observed the hydrogels with kilopascal level modulus. In this aspect, low modulus of this material cannot be the main novelty of this work. In addition, in order to compare the modulus of the developed material and values in reported papers, the authors should measure the tensile properties since other factors, such as adhesiveness, inhomogeneity, microstructure, and surface roughness, can affect the data achieved by indenter.

6. We recommend the authors provide more adhesion test data of the developed material. The authors

have to test the adhesiveness of the material with phantom skin (both dry and wet), and other inorganic materials (for robust connection with electrode).

7. It is necessary to provide more data for the demonstration of injectable scenario. For example, the viscosity of the material, viscosity changes during formation, wetting of material with various substrates (including epicardium), flow on the dry and wet surfaces, thickness control, properties depending on the thickness (e.g. conductivity degradation time), adhesion, type of bonding between tissue and materials etc. should be demonstrated sophisticatedly.

8. (Page 6, line 146) Please provide more data regarding the characterization of the synthesized material, such as microstructure, crystallinity etc.

9. (Page 8, line 184) Please provide the data for the following sentence: "The application of proBICS to the excised heart and subsequent electrofunctionalization did not induce observable damage."

10. (Page 8, line 192) Please provide the relevant data for the following sentence: "Initially, hearts were beating at a frequency of 0.8 Hz. Applying voltage pulses stimulated eBICS to control the heart's beating rate, adapting to the stimulating frequency at 2 Hz. After stimulation, heartbeats returned to the original frequency of 0.8 Hz."

11. (Page 8, line 207) Please provide the relevant data for the following sentence: "Consistent with the ex vivo zebrafish heart experiment results, eBICS strongly adhered to the heart tissue in vivo."

Reviewer #2 (Remarks to the Author):

In this article, Aydemir et al. introduced a bioresorbable injectable cardiac stimulator (BICS), designed for minimally invasive heart stimulation without requiring surgical implantation or removal. The BICS system consists of a nanoparticle solution that, upon injection, formed a conductive network around the heart, enabling cardiac monitoring and arrhythmia correction. This system comprised a mixture of A5 (a PEDOT-S variant) and ETE-BuSA, which self-assembles into a stable conductive structure upon injection. Tested in zebrafish and chicken embryos, BICS demonstrated high biocompatibility, maintained conductivity for five days, and effectively regulated heartbeats without causing toxicity or long-term behavioral changes. The reviewer believes that this technology offers a potential alternative for temporary heart stimulation in remote or emergency scenarios and recommends its publication in Nature Communications after proper revision.

Comment #1: The author validated the properties of the hydrogel in zebrafish and ex vivo mouse-sized chicken embryo. However, in vivo demonstrations in mammalian models (e.g., rat or rabbit) should be additionally conducted to show potential of the hydrogel for clinical translation. Various experiments such as biocompatibility, functionality, and long-term stability should be included. Specifically, the practicality of the hydrogel should be proved through demonstration in moving animals, which would significantly enhance the relevance and impact of the research.

Comment #2: The authors claim that the BSIC adheres to the moving heart based on in vivo experiments. However, the current manuscript does not provide enough quantitative data on the adhesive properties of the hydrogel. The reviewer recommends conducting ex vivo experiments, such as lap shear tests and peel tests, to evaluate such key point of the hydrogel. A more detailed discussion of the chemistry about the hydrogel's adhesive properties would significantly enhance the quality of the manuscript. It would also be helpful to compare the adhesive properties of the new hydrogel with reported conductive hydrogels. This comparative analysis would provide a clearer picture of the progressiveness of the suggested technology.

Comment #3: Accurate fixation of the BSIC position at the target site is a major challenge, given its high solubility which allows it to disperse before aggregation in a wet in vivo environment. The reviewer suggests that the authors include data on the aggregation time for BSIC. Additionally, imaging data that shows precise administration of the BSIC might be helpful for supporting the author's claim.

Comment #4: The images in Figures 1 and 2 are not well-aligned and does not effectively deliver messages. Please revise images to present the novelty of the paper, such as minimally-invasive implantation of the hydrogel into the heart, biodegradability of the hydrogel, and conductivity of the hydrogel.

Comment #5: In the Abstract, the abbreviations PEDOT-S and ETE-BuSA are used without any explanation of what they represent. It is essential to include an explanation of these substances at the very beginning of the paper.

Comment #6: When 1V is applied to the BSIC for electrofunctionalization, it seems that the insulation might not be adequate. This raises the concern that 1V could also be delivered to adjacent skin, causing side effects. Please clarify this point and provide supporting evidence.

Comment #7: Please provide the signal-to-noise ratio for ECG recordings in Extended Data Fig. 4. The signal quality appears to deteriorate after day 4. Since the device is reported to be degraded on day 6, there are concerns about the feasibility of obtaining reliable ECG signals up to the day 5. Please clarify this discrepancy.

Comment #8: When using descriptive words such as "gross" and "elegant," please ensure their usage is appropriate to avoid any difficulty in interpretation. Please replace such terms to clear ones.

Comment #9: The following are sections that require additional explanation.

- On page 3, line 72, the term "injecting energy" is used. It is unclear what this term specifically refers to. Please provide a more detailed explanation.
- On page 3, line 78, being soluble is not a prerequisite for being injectable. Please clarify the intended meaning behind grouping these terms together to make it easier to understand. Alternatively, consider revising the terminology used.
- On page 4, lines 79-80, the term "duality" implies a trade-off between two paired attributes. However,

it is difficult to see adherence and elasticity as items in a trade-off relationship. Additional explanation is needed to clarify this point.

- On line 114 and line 118, it is mentioned that ETE-PC is dispersed within the A5 core, while ETE-BuSA is highly soluble in A5. However, further explanation is needed to clarify the structural differences between these two substances.

- Please ensure that appropriate explanations are added to the Extended Data as well.

Response to reviewers:

We thank the reviewers for carefully reading through and helping us to improve the manuscript.

REVIEWER COMMENTS

Reviewer #1 (Remarks to the Author):

This paper presents an injectable conductive hydrogel as an electrode for the cardiac application, including both ECG monitoring and electrical stimulation. The material consists of A5 (a variant of PEDOT-S) and ETE-Busa and has low Young's modulus (21kPa) with high conductivity (55 S/m). The authors demonstrate the efficacy of this material by using zebrafish and chicken embryo.

1. We recommend that the authors provide additional data to support their claim – (1) bioresorbability. Although the title of this manuscript is "bioresorbable injectable cardiac stimulator", data that support the bioresorbability of the materials is insufficient. The authors have to provide relevant data. In addition, thiophene, in its basic form, is not inherently bioresorbable due to their stable aromatic ring structure, and this is the reason why PEDOT series are known as biocompatible but non-degradable (and non-bioresorbable) materials. In principle, bioresorbable materials refer to the materials those can be broken down and absorbed by the body, usually through natural biological processes. In other words, if it is bioresorbable, it should be hydrolysable or can be broken by the enzyme in our body. Therefore, the authors should demonstrate the bioresorbability of the synthesized material.

Since there is no consensus on the terminology, and to maintain consistency with our previous publications, we have removed the term "bioresorbable" from the title but retained it in the main text.

Looking into the terminology, we follow the definition by John Rogers and Robert Langer and colleagues. Rogers <https://doi.org/10.1002/adma.202309421> has used this definition in publications in the Nature portfolio, e.g., <https://doi.org/10.1038/s41587-021-00948-x> and Science portfolio, DOI: 10.1126/sciadv.abp9169. Langer and colleagues state that there is no consensus on the terminology used (<https://doi.org/10.1016/B978-0-08-087780-8.00021-8>). For example, bioresorbable and bioabsorbable are used interchangeably; the only consensus is that biodegradable should include cleavage of a covalent bond with an enzyme, cell, or microorganism. Or so we thought until last week, Bao and coworkers recently published a review (<https://doi.org/10.1038/s44222-024-00194-1>) "*The term 'biodegradable' is often used interchangeably with 'bioresorbable' in the field of bioelectronics — the latter refers to complete degradation within the human body without leaving any remaining foreign material*". Thus, there is no to little consensus.

Rogers' use of silicon does not fit the criterion of Reviewer 1. In several other studies, alloys including Gallium <https://www.mdpi.com/2306-5354/10/2/273> and Molybdenum <https://www.sciencedirect.com/science/article/pii/S2452199X21005260>, have been defined as bioresorbable.

The general statement on the stability of thiophenes varies based on the context. Electron-rich thiophenes in drugs are extensively metabolized, particularly oxidized at the 1, 2-, and/or 5-positions, leading to their breakdown. For PEDOT derivatives, stability depends on their association, such as with PSS (or other doping molecules) in hydrogels or when used as metal coatings. Remarkably, little is known about PEDOT's in vivo metabolism despite its extensive use as a model for conductive polymers, with over 21,000 publications mentioning PEDOT in an in vivo context, according to a quick Google Scholar search.

Our system absorbs ions from the tissue to form the conducting structure, and these ions are subsequently resorbed during the disintegration process. Thus, parts of the eBICS electrode are bioresorbed according to all above definitions.

An extensive metabolism study of PEDOT-S compared to PEDOT:PSS should be highly interesting. However, it deserves its own attention and cannot be confined to the present study.

2. The authors should add more data regarding the biocompatibility and toxicity of the synthesized material and of its degradation byproduct. Histological examination of myocardial tissue near the site of material attachment, immunohistochemistry should be conducted.

In the manuscript, we present toxicity on the organism level (including offspring of treated animals), organ level (heart), and cellular level. Immunohistochemistry on the heart at the site of the material is included in Figure 4, and we have clarified this in the figure legend. We are also citing an earlier study from our group (<https://doi.org/10.1038/s41467-023-40175-3>) where A5 was injected into brains without causing adverse tissue response.

3. We recommend the authors revise the title since it can mislead the readers. What the authors developed is just

electrode, and it is not a cardiac stimulator. Cardiac stimulator generally refer to the pacemaker with battery and controlling system, but this system should be connected with external stimulator (or ECG sensor) to be operated.

We follow the terminology by John A. Rogers, <https://doi.org/10.1038/s41587-021-00948-x>, that also uses an external stimulator to complete the fully implantable and resorbable pacemaker.

It is recognized that the design of in-vivo electrodes and their introduction into the body is the most complicated part of the device; this goes for pacemakers, neurostimulators (NeuroLink), and DBS/Pain devices. The external electronics for these devices are well-developed; in our case, the end game is a mobile app.

4. The authors should add more information to support their claim – (2) the necessity of the higher conductivity of the electrode. As the authors mentioned, injectable hydrogel is already demonstrated in many papers, and they show reliable stimulation (or pacing) performance. The authors should explain the advantages of higher conductivity and demonstrate it in the main manuscript with comparison data. It is also recommended to provide a table that compares the various properties, including electrical conductivity, adhesiveness, degradability, viscosity etc., of published hydrogels and developed material.

As we write in the paper, all injectable conductive hydrogels reported are used in the passive form as patches or bridges. Thus, they are not externally connected and neither record nor stimulate heartbeats. Higher conductivity is needed to stimulate without increasing the supplied voltage, such as when, for example, potentially using a mobile phone.

We note that Jin et al. recently published a paper on injectable tissue prostheses in Nature (Nature volume 623, pages 58–65 (2023)). One of the key enablers for their study was the discovery of a conductive hydrogel that could be injected using small-diameter capillaries. In Supplemental Figure 17, they surveyed the field of such published hydrogels and the injection capillaries used to inject them. We plotted the values from their study and added our own data point; see Figure below. The graph shows that the proBICS/eBICS system vastly outperforms the previously published work on electrical conductivity and injectability, where a small inner diameter is sought for minimal invasiveness. The higher conductivity of the BICS allows us to contact it and use it in an active mode externally.

Figure. A comparison of the injectability and electrical conductivity of the proBICS/eBICS to other hydrogels was previously reported. Please note the logarithmic scales. Higher conductivity and smaller inner diameter are better. Modified from Jin et al., Nature volume 623, pages 58–65 (2023).

5. The authors need to do more literature review for the modulus of reported hydrogels. Although the authors highlight the low modulus of the sample with that of PEDOT:PSS hydrogel (Page 7, line 174), it is commonly observed the hydrogels with kilopascal level modulus. In this aspect, low modulus of this material cannot be the main novelty of this work. In addition, in order to compare the modulus of the developed material and values in reported papers, the authors should measure the tensile properties since other factors, such as adhesiveness, inhomogeneity, microstructure, and surface roughness, can affect the data achieved by indenter.

Yes, that is correct, but it is not the low modulus alone; the combination of high conductivity, low modulus, and injectability is one of the novelties. It is quite easy to do either, gold electrodes/substrate bound have high conductivities and high modulus, and blood has low modulus and low conductivity 14 mS/cm^{-1} (seawater has about the same conductivity as blood), the latter being similar to the reported conductive hydrogels (patches and bridges) presented in the graph for Reviewer 1 Q4).

Regarding tensile testing: we comprehensively answer about mechanical testing under Q6.

6. We recommend the authors provide more adhesion test data of the developed material. The authors have to test the adhesiveness of the material with phantom skin (both dry and wet), and other inorganic materials (for robust connection with electrode).

The eBICS electrode differs from implanted ex-vivo assembled bioelectronics in that it infiltrates the tissue both from the liquid injection and from the electrofunctionalization process (see, e.g., Figure 1b). Pull-off and lap-shear tests, which investigate the interface between the polymer and the underlying tissue, are therefore poorly suited to studying eBICS adhesion/infiltration. We welcome any collaborator to investigate the adhesion properties on phantom skin, but we prefer to keep this manuscript focused on applications relating to the heart.

Instead, we have added substantial qualitative and quantitative data on eBICS adhesion to heart tissue to the supplemental information. Pull-off force and surface energy were quantified using a microindenter, and we found that they were similar for the heart and the eBICS installed on the heart, being around 60 mJ/m^2 . From a qualitative view, we also added 3 movies and 2 figures to the supplemental information where we put eBICS-hearts under significant stress without being able to rinse or pull off the eBICS. This is not surprising since we already observed that the eBICS is functional for 5 days in an in-vivo setting. The zebrafish heart continuously beats with a rate of 120–180 Hz when not under anesthesia, corresponding to more than 50 million heart contractions under which the eBICS remained adhered, continuous, and functional within the fish.

7. It is necessary to provide more data for the demonstration of the injectable scenario. For example, the viscosity of the material, viscosity changes during formation, wetting of material with various substrates (including epicardium), flow on the dry and wet surfaces, thickness control, properties depending on the thickness (e.g. conductivity degradation time), adhesion, type of bonding between tissue and materials etc. should be demonstrated sophisticatedly.

We agree that these are important points to guide the design to be able to use small diameter injection capillaries. However, as we described in the Figure above, we already use an order of magnitude smaller diameter than competing technologies highlighting the high injectability of the proBICS. During electrofunctionalization, BICS infiltrates surrounding tissue/media making it very challenging to track viscosity changes during formation. From an application driven perspective, the most important aspects are the injectability using small diameter capillaries and the tissue matched mechanical properties post functionalization. Both of which are described in the manuscript.

In addition, we have added a figure to the supplemental information, Extended Figure 13, depicting proBICS wetting on various substrates. By carefully pipetting proBICS on 2D surfaces (Au and Si) as well as excised, beating chicken hearts, we could extract contact angles for the ability of the polymer to wet these surfaces which have vastly different chemistry. On both 2D surfaces, proBICS formed well-defined droplets with 122° contact angle. On the chicken heart, the polymer quickly spread along the surface and did not form a well-defined droplet thereby highlighting the ability of this solution to wet relevant tissue.

8. (Page 6, line 146) Please provide more data regarding the characterization of the synthesized material, such as microstructure, crystallinity etc.

The materials are extensively described: A5 in one of the references (<https://doi.org/10.1021/acs.chemmater.1c04342>) and ETE-BuSa in the manuscript.

We have performed additional experiments to characterize the eBICS which goes beyond what was described in earlier publications. By electrofunctionalizing the proBICS solution on a silicon wafer, we were able to image the dendritic structures down to the nanometer scale. Similarly to how it appears when injected into agarose and monitored using light microscopy, Figure 1c, we observe continuous dendritic structures consisting of ETE-BuSA functionalized onto and extending outwards from the A5 core. At high magnification, faceting can be observed, indicating some degree of micro crystallinity. We have added this data to the supplemental information file, Extended Figure 12.

9. (Page 8, line 184) Please provide the data for the following sentence: "The application of proBICS to the excised heart and subsequent electrofunctionalization did not induce observable damage."

We have added Extended Figure 15 to the supplemental information, which shows a zebrafish heart after eBICS installation.

10. (Page 8, line 192) Please provide the relevant data for the following sentence: "Initially, hearts were beating at a frequency of 0.8 Hz. Applying voltage pulses stimulated eBICS to control the heart's beating rate, adapting to the stimulating frequency at 2 Hz. After stimulation, heartbeats returned to the original frequency of 0.8 Hz."

We have added a figure to the supplemental information, Extended Figure 14, depicting ECG obtained from an eBICS-zebrafish heart before, during, and after stimulation.

11. (Page 8, line 207) Please provide the relevant data for the following sentence: "Consistent with the ex vivo zebrafish heart experiment results, eBICS strongly adhered to the heart tissue in vivo."

This was deduced by an ocular inspection. We have now revised the sentence to say "Consistent with the ex vivo zebrafish heart experiment results, eBICS adhered to the heart tissue in vivo and could not be rinsed away."

We have also added quantitative and qualitative data on eBICS adhesion to the supplemental information. See the answer above.

Reviewer #2 (Remarks to the Author):

In this article, Aydemir et al. introduced a bioresorbable injectable cardiac stimulator (BICS), designed for minimally invasive heart stimulation without requiring surgical implantation or removal. The BICS system consists of a nanoparticle solution that, upon injection, formed a conductive network around the heart, enabling cardiac monitoring and arrhythmia correction. This system comprised a mixture of A5 (a PEDOT-S variant) and ETE-BuSA, which self-assembles into a stable conductive structure upon injection. Tested in zebrafish and chicken embryos, BICS demonstrated high biocompatibility, maintained conductivity for five days, and effectively regulated heartbeats without causing toxicity or long-term behavioral changes. The reviewer believes that this technology offers a potential alternative for temporary heart stimulation in remote or emergency scenarios and recommends its publication in Nature Communications after proper revision.

We thank the reviewer for acknowledging the usefulness of the eBICS and the manuscript's suitability for Nature Communications.

Comment #1: The author validated the properties of the hydrogel in zebrafish and ex vivo mouse-sized chicken embryo. However, in vivo demonstrations in mammalian models (e.g., rat or rabbit) should be additionally conducted to show potential of the hydrogel for clinical translation. Various experiments such as biocompatibility, functionality, and long-term stability should be included. Specifically, the practicality of the hydrogel should be proved through demonstration in moving animals, which would significantly enhance the relevance and impact of the research.

The request was overruled by the editorial team. However, we would like to point out that the zebrafish were left to swim around for several days with functional eBICS installed. Hence, we have already proven the useability in moving animals.

Comment #2: The authors claim that the BSIC adheres to the moving heart based on in vivo experiments. However, the current manuscript does not provide enough quantitative data on the adhesive properties of the hydrogel. The reviewer recommends conducting ex vivo experiments, such as lap shear tests and peel tests, to evaluate such key point of the hydrogel. A more detailed discussion of the chemistry about the hydrogel's adhesive properties would significantly enhance the quality of the manuscript. It would also be helpful to compare the adhesive properties of the new hydrogel with reported conductive hydrogels. This comparative analysis would provide a clearer picture of the progressiveness of the suggested technology.

Please see our response to Reviewer 1's comment related to adhesion. In short, the eBICS infiltrates tissue and can not be regarded as a simple 2D material put on top. We have added quantitative data on pull-off force and surface energy and qualitative data when putting the heart/eBICS under significant stress.

Comment #3: Accurate fixation of the BSIC position at the target site is a major challenge, given its high solubility which allows it to disperse before aggregation in a wet in vivo environment. The reviewer suggests that the authors include data on the aggregation time for BSIC. Additionally, imaging data that shows precise administration of the BSIC might be helpful for supporting the author's claim.

The aggregation of these mixtures depends on the ion concentrations in the tissue and the A5 and specific ETE derivatives used. We have thoroughly investigated this both in vitro and in vivo, as detailed in the manuscripts (<https://doi.org/10.1021/acs.chemmater.1c04342> and <https://doi.org/10.1038/s41467-023-40175-3>). In this manuscript, we describe how the mixture of ETE-BuSA and A5 forms BICS instantly around the beating heart. For example, as described in the manuscript, the former used ETE-PC disperses the initial formation of the A5 structure.

We have also added a figure to the supplemental information, Extended Figure 16, showing a fish during the proBICS injection to give a clearer picture of the injection process.

Comment #4: The images in Figures 1 and 2 are not well-aligned and does not effectively deliver messages. Please revise images to present the novelty of the paper, such as minimally-invasive implantation of the hydrogel into the heart, biodegradability of the hydrogel, and conductivity of the hydrogel.

This was a mistake on our end; we have now aligned the pictures and graphs in Figure 1 and Figure 2.

Comment #5: In the Abstract, the abbreviations PEDOT-S and ETE-BuSA are used without any explanation of what they represent. It is essential to include an explanation of these substances at the very beginning of the paper.

Yes, we will add that to the abstract.

Comment #6: When 1V is applied to the BSIC for electrofunctionalization, it seems that the insulation might not be adequate. This raises the concern that 1V could also be delivered to adjacent skin, causing side effects. Please clarify this point and provide supporting evidence.

We have not seen any damage to the skin or electrode surrounding tissue neither in the making of this manuscript, nor in a previous one where we installed electrodes in notoriously sensitive brain tissue (<https://doi.org/10.1038/s41467-023-40175-3>). High electric fields, such as those used when doing electroporation (doi: 10.1242/dmm.034561), are known to induce tissue damage, but the voltage we use is about 1/50th of what is used in such applications, further corroborating the lack of skin damage observed in our setup. Additionally, Rodney P O'Connor recently reported that PEDOT:PSS coated electrodes reduce cell damage and intracellular oxidation (<https://doi.org/10.1016/j.bioelechem.2022.108163>). Only cells within the distance of 10–60 μm were affected during electroporation, and the field was negligible at more than 100 μm from the electrode 20 V pulses <https://theses.hal.science/tel-03845033>.

Relating to the intended use, a possible minimal effect on the skin would be a reasonable trade-off, as the alternative could be fatal. However, if there is a problem, injection through a thin patch of insulation material to protect the skin would be a straightforward and viable solution.

Comment #7: Please provide the signal-to-noise ratio for ECG recordings in Extended Data Fig. 4. The signal quality appears to deteriorate after day 4. Since the device is reported to be degraded on day 6, there are concerns about the feasibility of obtaining reliable ECG signals up to the day 5. Please clarify this discrepancy.

The bioresorption of the BICS is a process where the electrode continuously degrade and is fully bioresorbed after 1 week. We have visually observed that the degradation process is accelerated during the last days. In addition to the in vivo bioresorption, the eBICS patch on top of the zebrafish skin is also removed, assisted by the hydrodynamic motion arising when the fish are swimming around. We observed ECG S/N ratios for the 2nd, 3rd, 4th, and the 5th day. 14.2, 14.6, 14.4, and 13.9 dB. We defined a meaningful signal to be a signal that shows that the heart is beating. After day 5, we were unable to record meaningful signals.

Comment #8: When using descriptive words such as "gross" and "elegant," please ensure their usage is appropriate to avoid any difficulty in interpretation. Please replace such terms to clear ones.

Yes, agree; we have removed gross and elegant as they do not add anything to the manuscript.

Comment #9: The following are sections that require additional explanation.

- On page 3, line 72, the term "injecting energy" is used. It is unclear what this term specifically refers to. Please provide a more detailed explanation.

The sentence in the manuscript is as follows: "In addition to injecting enough energy for cardiac stimulation", and not "injecting energy". However, to clarify the sentence, we have changed it to: "In addition to applying a high enough bioelectric field for cardiac stimulation"

- On page 3, line 78, being soluble is not a prerequisite for being injectable. Please clarify the intended meaning behind grouping these terms together to make it easier to understand. Alternatively, consider revising the terminology used.

We agree that solubility is not a prerequisite for being injectable; it depends on the context. The sentence is written as follows: "Thus, the formulation must be highly soluble for injection but aggregate into a highly conducting structure in vivo to adhere to the beating heart and form an external connection"

Solubility and aggregations are connected; the need for highly soluble materials implies having enough material in high concentration to generate a conductive structure. Solubility is necessary in the first place so that it can template the heart's structure when it aggregates.

- On page 4, lines 79-80, the term "duality" implies a trade-off between two paired attributes. However, it is difficult to see adherence and elasticity as items in a trade-off relationship. Additional explanation is needed to clarify this point.

Duality does not necessarily imply a meaning of trade-off. The definition "Duality generally refers to the idea that two opposite properties are connected"

- On line 114 and line 118, it is mentioned that ETE-PC is dispersed within the A5 core, while ETE-BuSA is highly soluble in A5. However, further explanation is needed to clarify the structural differences between these two substances.

ETE-PC: Phosphatidylcholine (PC) is a zwitterionic molecule on cell membranes. It likely evolved to prevent unwanted interactions with proteins and other molecules, both inside and outside the cell, thanks to its antifouling properties. Zwitterions, like PC, are commonly used for their ability to reduce such interactions. The antifouling effect of ETE-PC also disrupts the initial aggregation effect on A5 on the heart surface.

That is what the sentence in the manuscript describes: ...ETE-S and ETE-PC, which did not generate stable structures on the heart when combined with A5. In fact, the zwitterionic trimer ETE-PC dispersed the A5 core structure before electrofunctionalization.

ETE-BuSA: BuSA is also a zwitterionic molecule. However, unlike previously described sulfonate amine zwitterions, ETE-BuSA has an extra carbon between the negative sulfonate group and the positive amine group. We designed BuSA as a zwitterion with reduced antifouling properties, but it still has a positive amine group to interact with negatively charged cell surfaces. Additionally, it maintains high solubility when combined with A5, which ETE derivatives that only contain positive-charged amines do not.

The sentence in the manuscript: The mixture of ETE-BuSA and A5 (proBICS) was highly soluble.

- Please ensure that appropriate explanations are added to the Extended Data as well.

We have added several supplemental movies and figures with descriptions.

REVIEWERS' COMMENTS

Reviewer #1 (Remarks to the Author):

1.

First of all, the consensus of the terminology – bioresorbable - is not the main issue. Since the authors stated that bioresorbability is one of the main novelties of this paper, the authors should provide proper data to show the bioresorbable or biodegradable behavior of the developed material. However, according to the statement in the response to reviewer (“Thus, parts of the eBICS electrode are bioresorbed according to all above definitions.”), this material cannot be defined as a “bioresorbable (or biodegradable)” material. Then, it reduces the novelty of this manuscript.

Second, the reviewer cannot agree with the following statement - “Rogers' use of silicon does not fit the criterion of Reviewer 1”. Rogers group demonstrated the hydrolysable behavior of silicon (<https://www.science.org/doi/10.1126/science.1226325>; <https://pubs.acs.org/doi/10.1021/nn500847g>), and hydrolysable means the silicon can be bioresorbable as well as biodegradable, as long as it is the condition with water. This is the reason why Rogers group used the term “bioresorbable” in many of his publications.

Third, as the authors mentioned (“Remarkably, little is known about PEDOT's in vivo metabolism despite its extensive use as a model for conductive polymers, with over 21,000 publications mentioning PEDOT in an in vivo context, according to a quick Google Scholar search.”), in vivo metabolism of PEDOT is still unknown, and this is the reason why many researchers, including Zhenan Bao, does not claim that PEDOT as a bioresorbable (or biodegradable) material. However, in this manuscript, the authors claim that this material is a bioresorbable (or biodegradable) material without any supporting data or references, it is newly developed material though.

Again, if the authors cannot provide the proper data that show the degradable mechanism of the developed material in water (hydrolysable behavior) or by enzyme, it means that the bioresorbability (or biodegradability) of the developed material is not fully defined. Then, it reduces the main novelty of this manuscript.

2.

The device in the cited reference (<https://doi.org/10.1038/s41587-021-00948-x>) includes the silicon-based PIN diode as a rectifier, and without this electronic component, the device cannot be operated wirelessly. In other words, the reason why the device in the reference can be defined as a “cardiac stimulator” is not because of the external stimulator but the design of the device itself. In this context, again, the suggested device in this manuscript cannot be defined as a stimulator since it is impossible to fabricate the rectifier for the wireless energy communication using the proposed electrode only.

3.

Again, the reviewers recommend the authors demonstrating the viscoelastic behavior to demonstrate the novelty (“injectability”) of the developed material and compared it with other published data of hydrogel.

Reviewer #2 (Remarks to the Author):

The reviewer has no further comment and the revised manuscript is now ready for publication.

Response to reviewers

We thank the reviewers for their help in perfecting this manuscript and making the message more clear to the future readers.

REVIEWERS' COMMENTS

Reviewer #1 (Remarks to the Author):

1.

First of all, the consensus of the terminology – bioresorbable - is not the main issue. Since the authors stated that bioresorbability is one of the main novelties of this paper, the authors should provide proper data to show the bioresorbable or biodegradable behavior of the developed material. However, according to the statement in the response to reviewer (“Thus, parts of the eBICS electrode are bioresorbed according to all above definitions.”), this material cannot be defined as a “bioresorbable (or biodegradable)” material. Then, it reduces the novelty of this manuscript.

Second, the reviewer cannot agree with the following statement - “Rogers' use of silicon does not fit the criterion of Reviewer 1”. Rogers group demonstrated the hydrolysable behavior of silicon (<https://www.science.org/doi/10.1126/science.1226325>; <https://pubs.acs.org/doi/10.1021/nn500847g>), and hydrolysable means the silicon can be bioresorbable as well as biodegradable, as long as it is the condition with water. This is the reason why Rogers group used the term “bioresorbable” in many of his publications.

Third, as the authors mentioned (“Remarkably, little is known about PEDOT's in vivo metabolism despite its extensive use as a model for conductive polymers, with over 21,000 publications mentioning PEDOT in an in vivo context, according to a quick Google Scholar search.”), in vivo metabolism of PEDOT is still unknown, and this is the reason why many researchers, including Zhenan Bao, does not claim that PEDOT as a bioresorbable (or biodegradable) material. However, in this manuscript, the authors claim that this material is a bioresorbable (or biodegradable) material without any supporting data or references, it is newly developed material though.

Again, if the authors cannot provide the proper data that show the degradable mechanism of the developed material in water (hydrolysable behavior) or by enzyme, it means that the bioresorbability (or biodegradability) of the developed material is not fully defined. Then, it reduces the main novelty of this manuscript.

A: In the manuscript, we present a representative image where eBICS installed on a zebrafish heart has been completely cleared after 1 week (Figure 4b). We have thus shown that the eBICS is cleared from the body, but the detailed mechanism remains unknown.

Hydrolysis or enzymatic breakdown are possible, but there are other options as well: the A5 as well as the trimers are small (in difference to the large and commonly used PSS) and could be solubilized without any chemical reactions. Delving deeper into this topic is beyond the scope of the present manuscript and we prefer not to speculate further on this topic.

In summary, we have shown that the eBICS is cleared (bioresorbed/biodegraded) from the body, but not through which detailed mechanism. The novelty will thus remain high. We have now added a statement where we state that we do not know the mechanism behind the bioresorption.

2.

The device in the cited reference (<https://doi.org/10.1038/s41587-021-00948-x>) includes the silicon-based PIN diode as a rectifier, and without this electronic component, the device cannot be operated wirelessly. In other words, the reason why the device in the reference can be defined as a “cardiac stimulator” is not because of the external stimulator but the design of the device itself. In this context, again, the suggested device in this manuscript cannot be defined as a stimulator since it is impossible to fabricate the rectifier for the wireless energy communication using the proposed electrode only.

A: In our manuscript we solve the connection challenge by making a conductive eBICS track all the way up to the skin where we deposit an eBICS patch. In the Si based approach, this was not an option since a conductive lead would disrupt biological functions and risk infections. Instead, they had to resort to wireless communication of the protocol governing the stimulation pulses. At the end of the day, both approaches are similar in the need for an external pulse source and should thus carry similar naming conventions.

3.

Again, the reviewers recommend the authors demonstrating the viscoelastic behavior to demonstrate the novelty (“injectability”) of the developed material and compared it with other published data of hydrogel.

A: Here, it is important to stress that our results are experimental in nature. We have installed eBICS electrodes in numerous fish with small diameter (30 μm) capillaries. Detailing the viscoelastic behavior would be excellent to guide the development if we *could not* use these very small capillaries (which we can).

Reviewer #2 (Remarks to the Author):

The reviewer has no further comment and the revised manuscript is now ready for publication.

A: We thank the reviewer for this positive feedback.